# Impulse Response Functions for Nonlinear, Nonstationary, and Heterogeneous Systems, Estimated by Deconvolution and Demixing of Noisy Time Series

**DOI:** 10.3390/s22093291

**Published:** 2022-04-25

**Authors:** James W. Kirchner

**Affiliations:** 1Department of Environmental Systems Science, ETH Zurich, CH-8092 Zürich, Switzerland; kirchner@ethz.ch; 2Swiss Federal Research Institute WSL, CH-8903 Birmensdorf, Switzerland; 3Department of Earth and Planetary Science, University of California, Berkeley, CA 94720-4767, USA

**Keywords:** impulse response function, deconvolution, system identification, ARMA noise, nonlinear deconvolution, robust estimation, time series analysis

## Abstract

Impulse response functions (IRFs) are useful for characterizing systems’ dynamic behavior and gaining insight into their underlying processes, based on sensor data streams of their inputs and outputs. However, current IRF estimation methods typically require restrictive assumptions that are rarely met in practice, including that the underlying system is homogeneous, linear, and stationary, and that any noise is well behaved. Here, I present data-driven, model-independent, nonparametric IRF estimation methods that relax these assumptions, and thus expand the applicability of IRFs in real-world systems. These methods can accurately and efficiently deconvolve IRFs from signals that are substantially contaminated by autoregressive moving average (ARMA) noise or nonstationary ARIMA noise. They can also simultaneously deconvolve and demix the impulse responses of individual components of heterogeneous systems, based on their combined output (without needing to know the outputs of the individual components). This deconvolution–demixing approach can be extended to characterize nonstationary coupling between inputs and outputs, even if the system’s impulse response changes so rapidly that different impulse responses overlap one another. These techniques can also be extended to estimate IRFs for nonlinear systems in which different input intensities yield impulse responses with different shapes and amplitudes, which are then overprinted on one another in the output. I further show how one can efficiently quantify multiscale impulse responses using piecewise linear IRFs defined at unevenly spaced lags. All of these methods are implemented in an R script that can efficiently estimate IRFs over hundreds of lags, from noisy time series of thousands or even millions of time steps.

## 1. Introduction

Many scientific challenges require understanding how systems respond to fluctuations in their inputs, as documented by sensor data streams or other time series data. Three examples from environmental science, for example, are (a) understanding how streamflow time series reflect landscape-scale responses to precipitation inputs, (b) quantifying source–receiver relationships for airborne or waterborne contaminants and pathogens, and (c) estimating how ecosystems’ rates of photosynthesis, respiration, and transpiration fluctuate in response to external forcings, such as solar radiation, precipitation, and nutrient concentrations. Many other examples can be found in fields ranging from chemistry [1], engineering [2], and economics [3,4] to systems biology [5], neuroscience [6,7], physiology [8] and epidemiology [9,10,11].

Questions such as these are both scientifically challenging and practically important. They can sometimes be investigated using controlled experiments, but it is often infeasible to conduct such experiments at realistic scale on systems of realistic complexity. Instead, we must often try to understand these processes by observing how systems respond, over time, to natural fluctuations in their external forcing and natural variations in their internal conditions. These efforts have been aided in recent decades by an ever-expanding range of observational time series from an increasingly diverse array of sensors and remote sensing platforms.

The question remains how to most usefully extract information from these signals. One widely employed approach is to postulate a model based on a set of assumed mechanisms, calibrate this model to the observational data, and then perform numerical experiments to explore the behavior of the model, under the assumption that the model is a realistic analogue to the real-world system. The strength of this approach is that results from these numerical experiments can be directly interpreted in terms of the postulated model processes. The corresponding weakness is that everything depends on the assumption that the postulated processes are the correct ones. Many models are sufficiently complex that even if they give the right answers (in the sense of matching their calibration data, for example), it can be difficult to tell whether they are doing so for the right reasons, which is essential to drawing valid inferences from their results.

An alternative approach is to make minimal assumptions about the processes of interest and instead to empirically model the input–output behavior of the system directly from observational data. The obvious drawback of such “black box” approaches is that any mechanistic inferences derived from them will necessarily be tentative and indirect (although in ways that will often be obvious to users). The corresponding advantage is that it is not necessary to assume a mechanistic model whose realism may be difficult to verify. 

This model-independent, data-driven approach, commonly termed *system identification*, has evolved primarily within the field of industrial process control (e.g., [12,13,14]). Industrial process control problems typically enjoy several advantages over analyses of other types of systems. In such problems, one often has strong a priori information about the structure of the systems under study (since they are typically engineered), and one can frequently measure their response to controlled inputs (steps, pulses, sine waves, etc.). In analyses of many other real-world systems, by contrast, the system’s structure is often unknown (which is often why it is being studied in the first place), and we must work with whatever patterns of inputs and outputs nature gives us. Furthermore, the input and output time series are frequently much noisier than in typical industrial systems.

Here I present new methods for system identification, with a specific focus on impulse response functions and how they can be adapted to handle several challenges that often arise in real-world systems. Section 2.1 briefly introduces impulse response functions and their estimation from time series via least squares methods. Section 2.2 through Section 2.6 show how this approach can be adapted to account for the autoregressive moving average (ARMA) noise that often arises in real-world data. Section 2.5 and Section 2.7 present benchmark tests demonstrating that this approach accurately estimates impulse response functions and their uncertainties, even when confronted with time series that are badly contaminated by nonstationary ARIMA (AutoRegressive Integrated Moving Average) noise.

Many real-world systems exhibit substantial heterogeneity, such that inputs to different system compartments are processed differently, with the resulting signals being mixed together in the system output. Section 3 presents a demixing approach to estimating the impulse response functions of these multiple overlapping (and correlated) inputs. Section 3 also presents benchmark tests that demonstrate the potential of this demixing approach.

Conventional impulse response functions assume that the system’s response is both linear (that is, proportional to the input) and stationary (that is, independent of time). Real-world systems, by contrast, are often nonlinear and nonstationary. Therefore, Section 4 shows how the demixing approach of Section 3 can be adapted to characterize systems’ nonstationary behavior, and Section 5 shows how this approach can be further extended to create piecewise linear maps of systems’ nonlinear dependence on their inputs. Section 6 further shows how IRFs can be approximated by piecewise linear functions that are evaluated at a few unevenly spaced knots, rather than many evenly spaced lags. This permits the accurate estimation of multiscale IRFs that combine brief, sharp impulse responses and more persistent, delayed impulse responses.

The techniques presented here are implemented in an R script, IRFnnhs.R (for “Impulse Response Functions for nonlinear nonstationary and heterogeneous systems in R”), which is available along with scripts for each of the benchmark tests presented in the following sections (see data availability statement). The benchmark tests presented here are intentionally generic, without applications to specific fields. A subsequent paper will present a proof-of-concept application within my own field of hydrology, but the presentation here is generic to avoid the misconception that these techniques are restricted to hydrological applications.

## 2. Estimating Impulse Response Functions from Time Series Contaminated by Autoregressive and Nonstationary Noise

### 2.1. Impulse Response Functions

Many systems can be represented (at least approximately) as convolutions, in which the output y(t) depends on an input x(t−τ) over a (potentially infinite) range of past lag times τ, weighted by a lag function β(τ) that expresses the relative influence of the input at each lag:(1)$$y\left(t\right)=\int_{0}^{\infty }\beta \left(\tau \right)\;x\left(t-\tau \right)\;d\tau .$$

The lag function β(τ) is sometimes called a *convolution kernel*, *transfer function*, or *Green’s function*; it is also called an *impulse response function* (IRF) because it shows how the system output y(t) would respond to an input x(t) consisting of a single Dirac delta function pulse (i.e., an infinitely high, infinitely narrow pulse that integrates to 1). A system’s impulse response can be defined more generally as the change in the time evolution of its output y when a single Dirac pulse $\delta_{t^{\prime }}$ is added to its input x(t) at any given time t′, compared to the system’s behavior without the Dirac pulse: (2)$$\beta_{t^{\prime }}\left(\tau \right)=y\left(t^{\prime }+\tau \;|\;x\left(t\right)+\delta_{t^{\prime }}\right)-y\left(t^{\prime }+\tau \;|\;x\left(t\right)\right).$$

The input x(t) can itself be considered as a continuous series of appropriately scaled Dirac pulses, so if the impulse response is independent of the impulse time t′ (that is, if the impulse response is stationary), integrating Equation (2) over all t′≤t will lead directly to the convolution shown in Equation (1). 

In most practical cases, continuous functions such as those in Equation (1) are not directly observable, and instead are approximated by discrete time series of measurements. In such cases, Equation (1) is typically approximated by its discrete counterpart,
(3)$$y_{j}=\overset{m}{\underset{k=0}{\sum }}\beta_{k}\;x_{j-k},\;j=1\ldots n.$$

The discrete impulse response function in Equation (3) is sometimes termed a finite impulse response (FIR) model, because it quantifies the finite-duration system response to a finite-duration input pulse, in contrast to Equation (1), which quantifies the potentially infinite-duration system response to an infinitesimally short input pulse.

The impulse response function β(τ) or $\beta_{k}$ is useful in characterizing the system; indeed, it is a complete description of linear time-invariant systems such as Equations (1) and (3). A *linear* system responds proportionally to the input x(t), such that its response to the sum of two inputs $x_{1}\left(t\right)$ and $x_{2}\left(t\right)$ equals the sum of its responses to the two inputs individually; this is known as the principle of linear superposition. A *time-invariant* (or *stationary*) system responds identically to the same inputs occurring at different times (except, of course, that its response is time-shifted by the same amount as the time difference between the inputs). 

In contrast to such an idealized linear time-invariant system, many real-world systems are nonlinear, nonstationary, or both. In such cases, an impulse response function will not be a complete description of the system’s response but may still be a useful indicator of its average behavior. Precisely how the IRF averages such a system’s behavior will depend on the system characteristics and on how the IRF is estimated; this topic is explored further in Section 3.3, Section 3.4, Section 3.5 and Section 4.3 below. Furthermore, as described in Section 4 and Section 5 below, the simple linear time-invariant model in Equation (3) can be generalized to estimate how the IRF varies for different input intensities (thus quantitatively characterizing the nonlinearity of the system) and to estimate how the IRF varies for inputs occurring at different times (thus quantifying the system’s nonstationarity). 

Convolutions such as Equations (1) and (3) scramble the input time series x and the impulse response function β together to generate the output time series y. Deconvolution methods seek to invert this process, un-scrambling y to yield estimates of x (given β) or estimates of β (given x). The term *deconvolution* is often applied specifically to the inversion of Equation (1) or (3) to solve for the input time series x given the output time series y and the impulse response function β (a deconvolution of y by β). Solving instead for the impulse response function β given the input and output time series x and y is also, mathematically speaking, a deconvolution (in this case, a deconvolution of y by x), but is also often termed *system identification* [13], since β characterizes the behavior of the system linking the inputs and outputs. The classical approach to either type of deconvolution relied on Fourier transform methods, because convolution and deconvolution become simply multiplication and division in Fourier space. However, Fourier methods often yield unreliable results unless the underlying system is linear and time-invariant, and thus conforms closely to Equation (1) or (3), and unless the two “knowns” (y and x for system identification, or y and β for deconvolution of the input time series x) are virtually noise-free. These requirements are often violated by real-world systems. Instead, the system identification problem is frequently approached by considering Equation (3) as a multiple linear regression equation,
(4)$$y_{j}=\overset{m}{\underset{k=0}{\sum }}\beta_{k}\;x_{j,k}+\alpha +\varepsilon_{j},$$where the *k*th column of the matrix $x_{j,k}=x_{j-k}$ is $x_{j}$ lagged by k time steps. Equation (4) can be straightforwardly solved for the IRF coefficients $\beta_{k}$ using conventional least squares methods if the residual errors $\varepsilon_{j}$ are uncorrelated white noise. 

### 2.2. Estimating Impulse Response Functions in the Presence of ARMA Noise

Direct application of simple approaches such as Equation (4) to many real-world systems will be complicated by the fact that the residuals $\varepsilon_{j}$ often violate the white noise assumptions underlying conventional linear regression. Instead, the residuals are often serially correlated, sometimes quite strongly, over a range of time scales, leading to biased estimates of the $\beta_{k}$ coefficients and their uncertainties. The serial correlation in $\varepsilon_{j}$ can arise from many sources. Measurements of the output variable $y_{j}$ may be subject to serially correlated, or even nonstationary, errors. The input variable $x_{j}$ may also be subject to error (the so-called “error in variables” problem). Even if those input errors are not themselves serially correlated, they will nonetheless be reflected in serially correlated residuals $\varepsilon_{j}$ because any excess or missing $x_{j}$ will appear to be smoothed and lagged by the same convolution process that smooths and lags the (unknown) true inputs. Equation (4) may also be a stationary approximation to a nonstationary real-world system, or may have other structural problems, such as missing variables or incorrect functional relationships, that would be reflected in serially correlated variations in the residuals $\varepsilon_{j}$. Efficiently handling these serially correlated errors requires novel statistical methods. Although several approaches have been widely used to perform regression in the presence of serially correlated errors (e.g., [15,16,17]; see also Section 9.5 of [12]), deconvolution in the presence of such errors is potentially a more complex problem, because the output will contain serially correlated signals from both the errors and the real-world convolution process, which must somehow be distinguished from one another.

Whatever the origin of the serial correlation in the residuals, it can be simply and flexibly represented as an Autoregressive Moving Average (ARMA) process,
(5)$$\varepsilon_{j}=\phi_{1}\;\varepsilon_{j-1}+\phi_{2}\;\varepsilon_{j-2}+\ldots +\xi_{j}+\vartheta_{1}\;\xi_{j-1}+\vartheta_{2}\;\xi_{j-2}+\ldots ,$$
where the autoregressive coefficients ϕ express how the residual $\varepsilon_{j}$ depends on its own previous values, and the moving average coefficients ϑ express how the residual $\varepsilon_{j}$ depends on the previous values of a white noise process $\xi_{j}$. In many real-world cases, serially correlated errors can be summarized using just a few autoregressive coefficients ϕ and moving average coefficients ϑ. Moving-average processes that are invertible (as all real-world moving average processes should be) can be equivalently expressed as autoregressive processes (the duality principle: see Section 3.3.5 of [12]), meaning that any real-world ARMA process can be re-expressed as a purely autoregressive (AR) process of higher order,
(6)$$\varepsilon_{j}=\phi_{1}\;\varepsilon_{j-1}+\phi_{2}\;\varepsilon_{j-2}+\ldots +\phi_{h}\;\varepsilon_{j-h}+\xi_{j}\;.$$

In theory, the autoregressive order h corresponding to a moving average process can be infinite, but in practice Equation (6) will often entail only a few more AR coefficients ϕ than the corresponding ARMA process in Equation (5), with the higher-order terms dying away to practically zero. 

A conventional approach to solving regression problems such as Equation (4) with autoregressive errors such as those in Equation (6) proceeds as follows. Equation (6) is first rearranged to express the uncorrelated white noise error $\xi_{j}$ in terms of the AR coefficients ϕ and the lagged values of the correlated errors $\varepsilon_{j}$: (7)$$\xi_{j}=\phi_{0}\;\varepsilon_{j}-\phi_{1}\;\varepsilon_{j-1}-\phi_{2}\;\varepsilon_{j-2}-\ldots -\phi_{h}\;\varepsilon_{j-h},$$where $\phi_{0}$, which has a value of 1, has been included in the first term on the right-hand side to make the following equations more systematic. Writing lagged copies of the original regression equation (Equation (4)) for lags 0 through h and multiplying by the corresponding AR coefficients ϕ yields the following stack of equations: (8)$$\phi_{0}y_{j}=\phi_{0}\sum_{k=0}^{m}\beta_{k}x_{j-k}+\phi_{0}\alpha +\phi_{0}\varepsilon_{j}\;-\phi_{1}y_{j-1}=-\phi_{1}\sum_{k=0}^{m}\beta_{k}x_{j-k-1}-\phi_{1}\alpha -\phi_{1}\varepsilon_{j-1}\;-\phi_{2}y_{j-2}=-\phi_{2}\sum_{k=0}^{m}\beta_{k}x_{j-k-2}-\phi_{2}\alpha -\phi_{2}\varepsilon_{j-2}\;\vdots \;-\phi_{h}y_{j-h}=-\phi_{h}\sum_{k=0}^{m}\beta_{k}x_{j-k-h}-\phi_{h}\alpha -\phi_{h}\varepsilon_{j-h}$$

Readers will note that the error terms in each line of Equation (8) sum up to the right-hand side of Equation (7), so if all of these lines are added together, the combined error terms will equal the uncorrelated error $\xi_{j}$:(9)$$\phi_{0}y_{j}-\phi_{1}y_{j-1}-\phi_{2}y_{j-2}-\cdots -\phi_{h}y_{j-h}\;=\phi_{0}\sum_{k=0}^{m}\beta_{k}x_{j-k}-\phi_{1}\sum_{k=0}^{m}\beta_{k}x_{j-k-1}-\phi_{2}\sum_{k=0}^{m}\beta_{k}x_{j-k-2}-\cdots -\phi_{h}\sum_{k=0}^{m}\beta_{k}x_{j-k-h}\;+\left(\phi_{0},-,\phi_{1},-,\phi_{2},-,\cdots ,-,\phi_{h}\right)\alpha \;+\left(\phi_{0},\varepsilon_{j},-,\phi_{1},\varepsilon_{j-1},-,\phi_{2},\varepsilon_{j-2},-,\cdots ,-,\phi_{h},\varepsilon_{j-h}\right)$$where the last line equals the uncorrelated error $\xi_{j}$. This conventional approach then rewrites Equation (9) by transforming each of the variables to subtract the values that they inherit from previous time steps,
(10)$$\begin{array}{c} y_{j}^{*}=\phi_{0}y_{j}-\phi_{1}y_{j-1}-\phi_{2}\varepsilon_{j-2}-\cdots -\phi_{h}y_{j-h}\\ x_{j}^{*}=\phi_{0}x_{j}-\phi_{1}x_{j-1}-\phi_{2}x_{j-2}-\cdots -\phi_{h}x_{j-h}\\ \alpha^{*}=\left(\phi_{0},-,\phi_{1},-,\phi_{2},-,\cdots ,-,\phi_{h}\right)\alpha \\ \xi_{j}=\varepsilon_{0}x_{j}-\varepsilon_{1}x_{j-1}-\varepsilon_{2}x_{j-2}-\cdots -\varepsilon_{h}x_{j-h} \end{array}$$yielding
(11)$$y_{j}^{*}=\overset{m}{\underset{k=0}{\sum }}\beta_{k}\;x_{j-k}^{*}+\alpha^{*}+\xi_{j}.$$

Equation (11) is in the form of a linear regression equation like Equation (4), but in place of the autocorrelated error term $\varepsilon_{j}$ it instead has the white-noise error term $\xi_{j}$, and thus conforms to the assumptions underlying regression analysis. As written, however, Equation (11) is nonlinear in its parameters and thus cannot be solved by linear regression (because the AR coefficients $\phi_{1}\ldots \phi_{h}$ hidden within the $x_{j-k}^{*}$ are multiplied by the regression coefficients $\beta_{k}$, as shown in the second line of Equation (9)). Historically, problems of this type were solved using the Cochrane–Orcutt procedure [15], which alternately estimates the AR coefficients ϕ and the regression coefficients β, iterating these two steps until convergence, or the Hildreth-Lu procedure [16], which solves jointly for the AR coefficients ϕ and the regression coefficients β using nonlinear search techniques [17]. More recent approaches iteratively estimate the β’s using Generalized Least Squares and the ϕ’s using maximum likelihood or Restricted Maximum Likelihood (REML) methods (see Section 9.5 of [12]). Pre-programmed routines are also available, such as the R language’s arima function, which estimates the β’s and ϕ’s using nonlinear optimization methods. However, these approaches can become slow and memory-intensive for large problems, such as those that arise when long time series are used to estimate convolution kernels over many lags. The order of difficulty of the matrix operations required to solve Equation (4) or (11) scales as roughly $n\;{\left(1+m\right)}^{2}$ or ${\left(1+m\right)}^{3}$ depending on the relative sizes of n and m; this is further magnified when these operations are repeatedly iterated to search for optimal values of the autoregressive coefficients $\phi_{1}\ldots \phi_{h}$. In addition to this computational issue, the differencing procedure in Equation (10) may amplify any errors in the input variables relative to the true input values (particularly if the true inputs are less time-varying than their errors, and thus are attenuated more by Equation (10) than their errors are), thereby magnifying the “errors in variables” problem [18,19].

A further potentially serious concern is that the regression coefficients $\beta_{k}$ estimated from Equations (9)–(11) or from R’s arima function will artifactually converge toward 0 at lags approaching the largest modeled lag m, even if the real-world process linking y and x extends to lags well beyond m. Even worse, the uncertainty estimates for these $\beta_{k}$ will also be artifactually driven toward 0 at lags approaching m, potentially misleading users into placing exaggerated confidence in these misleading results. The benchmark tests in Figure 1 show that these artifacts can occur even when the correct AR coefficients ϕ are known exactly (which of course will not be true in real-world cases).

Here I present a somewhat different approach that can efficiently handle large system identification problems in the presence of ARMA noise, and that is not vulnerable to the artifactual behavior shown in Figure 1. This approach is based on the observation that because Equation (9) is a convolution, it can be transformed into a conventional multiple linear regression problem that can be solved for coefficients that combine both the β’s and the ϕ’s, and these coefficients can then be back-transformed to extract the desired β values. The key is to recognize that the terms in the second line of Equation (9) can be aligned as follows (showing the first three rows as an example):(12)$$\phi_{0}\beta_{0}x_{j}+\phi_{0}\beta_{1}x_{j-1}+\phi_{0}\beta_{2}x_{j-2}+\phi_{0}\beta_{3}x_{j-3}+\cdots +\phi_{0}\beta_{m}x_{j-m}\;-\phi_{1}\beta_{0}x_{j-1}-\phi_{1}\beta_{1}x_{j-2}-\phi_{1}\beta_{2}x_{j-3}-\cdots -\phi_{1}\beta_{m-1}x_{j-m}-\phi_{1}\beta_{m}x_{j-m-1}\;-\phi_{2}\beta_{0}x_{j-2}-\phi_{2}\beta_{1}x_{j-3}-\cdots -\phi_{2}\beta_{m-2}x_{j-m}-\phi_{2}\beta_{m-1}x_{j-m-1}-\phi_{2}\beta_{m}x_{j-m-2}$$
where, readers will recall, $\phi_{0}=1$ and thus can be included or excluded without loss of generality. The vertically aligned columns in Equation (12) show that collecting terms with the same lag in x will convert Equation (9) to
(13)$$\phi_{0}\;y_{j}-\phi_{1}\;y_{j-1}-\phi_{2}\;y_{j-2}-\ldots -\phi_{h}\;y_{j-h}=\overset{m+h}{\underset{k=0}{\sum }}b_{k}\;x_{j-k}+a+\xi_{j},.\;$$which can be rearranged to create a conventional linear regression equation,
(14)$$y_{j}=\overset{m+h}{\underset{k=0}{\sum }}b_{k}\;x_{j-k}+\phi_{1}\;y_{j-1}+\phi_{2}\;y_{j-2}+\ldots +\phi_{h}\;y_{j-h}+a+\xi_{j},.\;$$where the error term $\xi_{j}$ is white noise, and the coefficients $b_{0}\ldots b_{m+h}$ and $\phi_{1}\ldots \phi_{h}$. can be jointly estimated by least-squares regression, with the matrix form (here using m = 5 and h = 2 as a simple example): (15)$$\left[\begin{array}{c} y_{8}\\ y_{9}\\ y_{10}\\ y_{11}\\ y_{12}\\ y_{13}\\ y_{14}\\ y_{15}\\ y_{16}\\ \vdots \end{array}\right]=\left[\begin{array}{cccccccc} x_{8} & x_{7} & \cdots & x_{2} & x_{1} & y_{7} & y_{6} & 1\\ x_{9} & x_{8} & \cdots & x_{3} & x_{2} & y_{8} & y_{7} & 1\\ x_{10} & x_{9} & \cdots & x_{4} & x_{3} & y_{9} & y_{8} & 1\\ x_{11} & x_{10} & \cdots & x_{5} & x_{4} & y_{10} & y_{9} & 1\\ x_{12} & x_{11} & \cdots & x_{6} & x_{5} & y_{11} & y_{10} & 1\\ x_{13} & x_{12} & \cdots & x_{7} & x_{6} & y_{12} & y_{11} & 1\\ x_{14} & x_{13} & \cdots & x_{8} & x_{7} & y_{13} & y_{12} & 1\\ x_{15} & x_{14} & \cdots & x_{9} & x_{8} & y_{14} & y_{13} & 1\\ x_{16} & x_{15} & \cdots & x_{10} & x_{9} & y_{15} & y_{14} & 1\\ \vdots & \vdots & \vdots & \vdots & \vdots & \vdots & \vdots & \vdots \end{array}\right]\cdot \left[\begin{array}{c} b_{0}\\ b_{1}\\ b_{2}\\ b_{3}\\ b_{4}\\ b_{5}\\ b_{6}\\ b_{7}\\ \phi_{1}\\ \phi_{2}\\ \alpha \end{array}\right]$$
or equivalently
(16)y=X · θ ,where the matrix X includes 1+m+h columns with lagged values of x and a further h columns with lagged values of y, and the parameter vector θ includes both the b and ϕ coefficients. (In practice the first m+h rows of the X matrix must be omitted because they have missing values, along with the corresponding rows of y; any other rows with missing values in either X or y are similarly omitted.) The least-squares estimate of the parameter vector θ can be computed via the conventional matrix form of the “Normal Equation” of linear regression,
(17)$$\theta ={\left(X^{T}X\right)}^{-1}\;\cdot \;X^{T}y,$$where the superscript T indicates the matrix transpose. If individual rows of Equation (15) are given different weights (for example, to exclude or down-weight uncertain or irrelevant observations), Equation (17) becomes
(18)$$\theta ={\left(X^{T}WX\right)}^{-1}\;\cdot \;X^{T}Wy,$$where W is a diagonal matrix containing the weights. The effective sample size, accounting for the uneven weights $w_{ii}$, can be calculated straightforwardly as $n_{\mathrm{eff}}={\left(\sum w_{ii}\right)}^{2}/\sum \left(w_{ii}^{2}\;\right)$; this converges to n when all rows have the same weight. The IRF function in IRFnnhs.R computes θ using R’s solve function, which in turn calls LAPACK solver routines based on LU decomposition. This is more efficient than inverting the cross-product matrix $X^{T}X$ or $X^{T}WX$ followed by matrix multiplication with $X^{T}Wy$. For most problems where n is much larger than, the IRF function’s runtime is approximately linear in and m, because the most time-consuming step is the construction of the X matrix. For larger m, runtime becomes linear in n and quadratic in m, because the most time-consuming step is the computation of the cross-product $X^{T}X$. For very large m, the order of difficulty may approach $m^{3}$ if the most time-consuming step becomes solving the linear system. Solution times on a 2019-vintage 1.8 GHz Intel i7-8550 CPU with four cores and 16 GB of RAM are roughly 50 ms for an X matrix with 10,000 rows and 100 columns, roughly 5 s for an X matrix with 100,000 rows and 1000 columns, and roughly 27 min for an X matrix with 1,000,000 rows and 10,000 columns. For comparison, R’s built-in arima function takes roughly 2000 times longer to solve the smallest of these problems, and for larger problems the discrepancy is even greater.

Because the $b_{k}$ coefficients estimated as part of θ in Equation (17) or (18) could be noisy if the underlying time series are short or particularly noisy, the IRF function provides an option for Tikhonov–Phillips regularization, as described in Equations (46), (49) and (50) of [20]. This regularization routine minimizes the mean square of the second derivatives of the $b_{k}$, thus penalizing $b_{k}$ values that deviate greatly from a line connecting their adjacent neighbors. This smoothness criterion has the advantage that it does not create a downward bias in the $b_{k}$ values, as conventional Tikhonov “ridge regression” would (see Section 4.3 of [20] for details). The degree of regularization is controlled by a dimensionless parameter ν that ranges between 0 and 1 and expresses the fractional weight given to the regularization criterion, relative to the least-squares criterion, in determining the best-fit values of θ. The default value of ν = 0 (no regularization) is used for all of the analyses presented here. 

As with any least-squares multiple regression problem, Equations (17) and (18) are potentially vulnerable to outliers in the underlying input and output time series. Therefore, the IRFnnhs.R script includes the option for robust solution of Equations (17) and (18) via Iteratively Reweighted Least Squares (IRLS). This robust estimation method can be invoked by calling the IRF function with the robust option set to TRUE. Because it is an iterative algorithm, IRLS will increase the solution time, but usually only by small multiples. A potentially greater concern is that, as with any robust estimation method, there is always a risk of excluding valid data that just happen to have unusually large influence. Therefore, it is worthwhile to investigate further, whenever robust and non-robust methods yield substantially different results. In the analyses presented here, the robust option is kept at its default value of FALSE, because the synthetic benchmark data sets contain no outliers (although they do contain significant noise). 

The form of Equation (14) is similar to a conventional SISO (Single Input, Single Output) ARX (Autoregressive with eXogenous variables) model (e.g., [13,14]), but there are three essential differences. The first difference is that in ARX models, the focus is usually on the autoregressive part, and the objective is usually to be able to make one-step-ahead forecasts of the next value of $y_{j}$, based mostly on y’s relationship to its own prior values. By contrast, in the analysis presented here, the focus is on the exogenous variable x and its lags, and on estimating their structural relationship to y. 

The second difference is that in ARX models, the autoregressive terms $\phi_{1}\;y_{j-1}$, $\phi_{2}\;y_{j-2}$, etc., describe autoregressive behavior in the system itself (an “equation error model”), rather than correcting for autoregressive noise in the error term (an “output error model”). (Although these two model types can be combined in so-called CARARMA models, for which iterative and hierarchical estimation algorithms have been proposed e.g., [21], such complex models need not concern us here because in the present analysis only the noise is assumed to be autoregressive.) Because it attributes autoregressive behavior to $y_{j}$ rather than to $\varepsilon_{j}$, an ARX model evaluates the b coefficients only for lags from 0 to m rather than from 0 to m+h as shown in Equation (14). This distinction is important because without the extra coefficients $b_{m+1}\ldots b_{h}$, Equation (14) would not be the same as Equation (9) and thus a solution for Equation (14) would not be a solution for the original problem as specified by Equation (4) combined with Equation (6).

The third crucial difference is that in an ARX model, the b coefficients would directly measure the effects of the (lagged) external forcing $x_{j-k}$. Here, by contrast, these effects are measured by the β coefficients, which must be deconvolved from the b coefficients as described in the next section. 

### 2.3. Deconvolving the Impulse Response from the Fitted Coefficients

The impulse response coefficients β are not estimated by the b coefficients themselves, but rather by functions that combine the b coefficients and the AR coefficients. One can translate between the regression coefficients b and the impulse response coefficients β by recognizing from inspection of Equation (12) that the b coefficients are linear combinations of the impulse response coefficients β, weighted by the AR coefficients ϕ,
(19)$$\begin{array}{l} b_{0}=\phi_{0}\beta_{0}\\ b_{1}=\phi_{0}\beta_{1}-\phi_{1}\beta_{0}\\ b_{2}=\phi_{0}\beta_{2}-\phi_{1}\beta_{1}-\phi_{2}\beta_{0}\\ \cdots \\ b_{m}=\phi_{0}\beta_{m}-\phi_{1}\beta_{m-1}-\phi_{2}\beta_{m-1}-\cdots -\phi_{h}\beta_{m-h}\\ b_{m+1}=-\phi_{1}\beta_{m}-\phi_{2}\beta_{m-1}-\cdots -\phi_{h}\beta_{m-h+1}\\ \cdots \\ b_{m+h-2}=-\phi_{h-2}\beta_{m-2}-\phi_{h-1}\beta_{m-1}-\phi_{h}\beta_{m}\\ b_{m+h-1}=-\phi_{h-1}\beta_{m-1}-\phi_{h}\beta_{m}\\ b_{m+h}=-\phi_{h}\beta_{m} \end{array}$$

These relationships can be represented in matrix form as (again using m = 5 and h = 2 as a simple example)
(20)$$\left[\begin{array}{c} b_{0}\\ b_{1}\\ b_{2}\\ b_{3}\\ b_{4}\\ b_{5}\\ b_{6}\\ b_{7} \end{array}\right]=\left[\begin{array}{cccccccc} 1 & 0 & 0 & 0 & 0 & 0 & 0 & 0\\ -\phi_{1} & 1 & 0 & 0 & 0 & 0 & 0 & 0\\ -\phi_{2} & -\phi_{1} & 1 & 0 & 0 & 0 & 0 & 0\\ 0 & -\phi_{2} & -\phi_{1} & 1 & 0 & 0 & 0 & 0\\ 0 & 0 & -\phi_{2} & -\phi_{1} & 1 & 0 & 0 & 0\\ 0 & 0 & 0 & -\phi_{2} & -\phi_{1} & 1 & 0 & 0\\ 0 & 0 & 0 & 0 & -\phi_{2} & -\phi_{1} & 1 & 0\\ 0 & 0 & 0 & 0 & 0 & -\phi_{2} & -\phi_{1} & 1 \end{array}\right]\cdot \left[\begin{array}{c} \beta_{0}\\ \beta_{1}\\ \beta_{2}\\ \beta_{3}\\ \beta_{4}\\ \beta_{5}\\ 0\\ 0 \end{array}\right]$$or more compactly as
(21)b=Φ · β.where Φ is a unit lower triangular Toeplitz matrix whose off-diagonals are the *negatives* of $\phi_{1}\ldots \phi_{h}$. This mapping of *b* to β is invertible, so the impulse response coefficients $\beta_{0}\ldots \beta_{m}$ can be retrieved from regression estimates of $b_{0}\ldots b_{m+h}$ and $\phi_{1}\ldots \phi_{h}$ by inverting the system of linear equations in Equation (19). This inversion can be written as a series of simple recurrence relationships (remembering that the main diagonal of the Φ matrix is 1):(22)$$\begin{array}{l} \beta_{0}=b_{0}\\ \beta_{1}=b_{1}+\phi_{1}\beta_{0}\\ \beta_{2}=b_{2}+\phi_{1}\beta_{1}+\phi_{2}\beta_{0}\\ \cdots \\ \beta_{m}=b_{m}+\phi_{1}\beta_{m-1}+\phi_{2}\beta_{m-2}+\ldots +\phi_{h}\beta_{m-h}\\ \beta_{i}=b_{i}+\sum_{j=1}^{\min\left(i,h\right)}\beta_{i-j}\;\phi_{j} \end{array}.$$

In matrix notation, this inversion can be expressed as (here again using m = 5 as an example)
(23)$$\left[\begin{array}{c} \beta_{0}\\ \beta_{1}\\ \beta_{2}\\ \beta_{3}\\ \beta_{4}\\ \beta_{m} \end{array}\right]=\left[\begin{array}{cccccc} 1 & 0 & 0 & 0 & 0 & 0\\ \psi_{1} & 1 & 0 & 0 & 0 & 0\\ \psi_{2} & \psi_{1} & 1 & 0 & 0 & 0\\ \psi_{3} & \psi_{2} & \psi_{1} & 1 & 0 & 0\\ \psi_{4} & \psi_{3} & \psi_{2} & \psi_{1} & 1 & 0\\ \psi_{5} & \psi_{4} & \psi_{3} & \psi_{3} & \psi_{1} & 1 \end{array}\right]\cdot \left[\begin{array}{c} b_{0}\\ b_{1}\\ b_{2}\\ b_{3}\\ b_{4}\\ b_{m} \end{array}\right]$$or more compactly as
(24)β=Ψ · b,where Ψ is the inverse of the matrix Φ, and both Ψ and b are truncated at lag m. Readers may recognize Equation (20) as a convolution, and Equation (23) as the corresponding deconvolution. The Ψ matrix, like the Φ matrix, is a lower unit triangular Toeplitz matrix, and the individual ψ values can be calculated by recurrence relationships analogous to those in Equation (22) above (noting that the terms of Φ are $-\phi_{j}$, not $\phi_{j}$, and that Φ and Ψ both have 1s along the main diagonal):(25)$$\begin{array}{l} \psi_{0}=1\\ \psi_{1}=\phi_{1}\\ \psi_{2}=\psi_{1}\;\phi_{1}+\phi_{2}\;\\ \psi_{3}=\psi_{2}\;\phi_{1}+\psi_{1}\;\phi_{2}+\phi_{3}\\ \cdots \\ \psi_{i}=\sum_{j=0}^{i-1}\psi_{j}\;\phi_{j-i} \end{array}\;.$$

The last *h* elements of b have no effect on the calculated $\beta_{k}$ values (each of which depends only on values of $b_{j\leq k}$—see Equation (22)), but they nonetheless must be present when Equation (14) is fitted by least squares, because otherwise the estimates of $b_{0}$ through $b_{m}$ can be distorted by correlations that should be absorbed by the terms $b_{m+1}$ through $b_{m+h}$. Putting the same point differently, if Equation (14) is missing the last h elements of b, it will not be equivalent to Equation (9), leading to biased estimates of the resulting $\beta_{k}$ (unless, of course, the last h elements of b are all zero). 

The Ψ matrix is also the Jacobian of the system of equations that translates b into β (Equation (22)), and thus it can also be used to convert the covariance matrix $K_{bb}$ of b to the covariance matrix $K_{\beta \beta }$ of the impulse response vector β, using the matrix form of the conventional first-order, second-moment error propagation equation,
(26)$$K_{\beta \beta }=\Psi \;\;\cdot \;K_{bb}\;\cdot \;\Psi^{T}.$$

The square root of the diagonal of the covariance matrix $K_{\beta \beta }$ will then yield the standard errors of the impulse response coefficients $\beta_{k}$. These uncertainty estimates will be somewhat inflated if Equation (26) is applied directly to the covariance matrix $K_{bb}$ obtained from Equation (14), due to interactions between the b and ϕ coefficients. Statistically speaking, the ϕ coefficients are “nuisance parameters” in the sense that the goal is to determine the values of the β’s, but as part of this process the ϕ’s must also be estimated in order to account for serial correlation in the residuals. Benchmark tests show that the standard errors of the β’s can be more accurately estimated if the ϕ’s are treated as being fixed at their estimated values, rather than as uncertain parameters. This can be efficiently accomplished by removing the rows and columns that correspond to the lagged values of y (and thus correspond to the ϕ coefficients) from the cross-product matrix $X^{T}X$ after it has been used to solve Equation (14). This modified cross-product matrix is then used to calculate the covariances $K_{bb}$ of the $b_{k}$ via the standard formula $K_{bb}=\sigma_{\xi }^{2}\;{\left(X^{T}X\right)}^{-1}$, where $\sigma_{\xi }^{2}$ is the variance of the residuals of Equation (14). The $b_{k}$ and their covariances $K_{bb}$ are then translated into estimates of the $\beta_{k}$ and their covariances $K_{\beta \beta }$ using Equations (24) and (26). 

### 2.4. Choosing the Number of Autoregressive Correction Terms

A practical question will inevitably arise: how should users choose the correct order h for the AR correction terms $\phi_{1}\;y_{j-1}\ldots \;\phi_{h}\;y_{j-h}$? As noted in Section 2.2 above, these terms should be numerous enough to capture the effects of both autoregressive and moving average noise in the measured $y_{j}$. One approach is to manually select the order of AR correction by trial and error, by setting the parameter h and inspecting how different values of h affect the estimates of $\beta_{k}$ and the correlations in the residuals. Benchmark tests suggest that as long as h is much smaller than *m*, making h too big will only slightly alter the estimates of $\beta_{k}$ or their uncertainties, since the extra ϕ coefficients will typically be very small and thus will barely affect the solution of Equation (14). (It should be clear that Equations (5) and (6) are models of the noise $\varepsilon_{j}$, not AR or ARMA models of the underlying processes generating the system output. Thus, any additional uncertainty in the ϕ coefficients due to overfitting is unproblematic, because the goal is to estimate the impulse response coefficients $\beta_{k}$, not the ϕ coefficients describing the noise.) 

The order of autoregressive correction h can also be determined automatically. One might assume that the Akaike Information Criterion (AIC) or Bayesian Information Criterion (BIC) could be used to determine an optimal value of h, but AIC and BIC can only compare models that predict the same set of points $y_{j}$, and changing h alters the number of rows of Equation (15) with missing values of $x_{j-m-h}$ or $y_{j-h}$ (and therefore the number of $y_{j}$ values that must be excluded). A more fundamental problem is that even if AIC and BIC could be used, they would select a value of h that makes Equation (14) a good predictor of $y_{j}$ rather than a good estimator of $b_{k}$ and thus of $\beta_{k}$, which is the objective of this analysis. One should remember that in this analysis, the coefficients $\phi_{1}\ldots \;\phi_{h}$ are not themselves of interest, but are necessary to whiten the residuals, that is, to convert the serially correlated residuals $\varepsilon_{j}$ to nearly uncorrelated residuals $\xi_{j}$. With that objective in mind, by default the IRF algorithm will automatically find the smallest value of h that sufficiently whitens the residuals, using both a practical significance test and a statistical significance test. The practical significance test examines whether the absolute values of the autocorrelation and partial autocorrelation function (ACF and PACF) coefficients of the residuals $\xi_{j}$, for lags from 1 to log_10_(n), are less than a user-specified threshold ARlim. The default value of ARlim is 0.2, which corresponds to a maximum contribution of 0.2^2^ = 4% to the variance of the residuals $\xi_{j}$ due to serial correlations at any individual lag. The statistical significance test examines whether the absolute values of the ACF and PACF coefficients exceed a significance threshold of $1.96/\sqrt{n_{\mathrm{eff}}}$ (corresponding to a two-tailed significance level of p < 0.05) more often than would be expected by chance according to binomial statistics, where the threshold for “by chance” is a user-specified probability ARprob (default value 0.05). The IRF algorithm will automatically find the smallest value of h that passes either of these tests. The practical significance test is needed because in large samples, even trivially small ACF and PACF coefficients may still be statistically significant, triggering a pointless effort to make them smaller than they need to be. Conversely, the statistical significance test is needed because in small samples, even true white-noise processes may yield ACF and PACF coefficients that do not meet the practical significance threshold (if ARlim is less than $1.96/\sqrt{n_{\mathrm{eff}}}$), triggering a pointless effort to further whiten residuals that are already white. 

A further technical detail is that if (but only if) the true impulse response function converges to 0 well before the maximum lag m, one can obtain more accurate estimates of $\beta_{k}$ at lags approaching m, with correspondingly smaller uncertainties, by solving Equations (10) and (11) using the ϕ coefficients obtained from Equation (14) rather than using the $\beta_{k}$ and uncertainty estimates derived from Equation (14) itself. Users can choose between these two alternatives with the complete option in the IRF function. If complete=TRUE, then the $\beta_{k}$ and their uncertainties are re-calculated via Equations (10) and (11) using the ϕ coefficients obtained from Equation (14). This is functionally equivalent to the Cochrane–Orcutt procedure, but is much faster because it does not require iteratively solving Equations (10) and (11) to determine the ϕ coefficients. Users should keep in mind, however, that if the true impulse response function does not converge to zero well before the maximum lag m, setting complete to TRUE can lead to artifacts similar to those shown in Figure 1; thus, complete is set to FALSE by default. 

### 2.5. Benchmark Tests

A simple benchmark test of the approach outlined above can be performed by generating a synthetic random input time series $x_{j}$, j=1…n, and convolving it with a known convolution kernel to yield the hypothetical system’s true output $y_{\mathrm{true},\;j}$. A random ARMA error time series $y_{\mathrm{err},\;j}$, generated with R’s arima.sim function, is then added to $y_{\mathrm{true},\;j}$ to yield the hypothetical system’s measured output $y_{j}$. This measured output and the input series $x_{j}$ are then supplied to the IRF function, which estimates the impulse response function coefficients $\beta_{k}$ and their standard errors. One can then calculate the estimation error as the difference between $\beta_{k}$ and the known convolution kernel, and take the root-mean-square average of these estimation errors at each lag (RMSE) from many random iterations of the benchmark test. If this RMSE approximately equals the pooled (root mean square) average of the estimated standard errors over many random iterations, then the standard errors are reasonable estimates of the uncertainties in the $\beta_{k}$. One can similarly calculate the mean estimation bias as the arithmetic average of the estimation errors at each lag. If this average estimation error scales roughly as the pooled standard error divided by the square root of the number of random iterations of the benchmark test, then this demonstrates that the $\beta_{k}$ estimates are unbiased (that is, no more biased than one would expect by chance) and the standard error estimates are realistic. 

This simple rubric for benchmark testing encompasses many different possibilities, depending on the chosen shape of the “true” convolution kernel, the time series length n, the maximum lag m, the distribution and correlation structure of the input signal $x_{j}$, and the amplitude and ARMA parameters of the error time series $y_{\mathrm{err},\;j}$. Figure 2 shows only a few illustrative examples. In all of these cases, n is 10,000, m is 60, $x_{j}$ is synthetic Gaussian noise with a lag 1 serial correlation of 0.5, and the error time series $y_{\mathrm{err},\;j}$ is ARMA(1,2) noise with an AR coefficient of 0.95 and MA coefficients of −0.2 and +0.2. The error time series is rescaled so that its variance equals the variance of $y_{\mathrm{true},\;j}$; thus, the signal-to-noise ratio is 1, implying a much larger level of noise than is commonly used in such benchmark tests. 

The three columns of Figure 2 show benchmark test results for gamma-distributed convolution kernels with shape factors of 1, 2, and 4. The top row of panels shows that IRFs estimated from time series corrupted with ARMA noise (dark blue points) correspond closely to the benchmarks (gray lines). The second row of panels shows that the RMS average deviations from the benchmarks in 100 random realizations of each benchmark test (dark blue points) agree with the expected error (as quantified by the pooled standard error, shown by the light blue lines). The third row of panels shows that the IRFs are unbiased; the average deviations from the benchmarks in 100 random realizations of each benchmark test (dark blue points) lie within their 95% confidence intervals, indicating that they are no larger than would be expected by chance in unbiased IRFs. The bottom row of panels shows 5% of each convolved time series $y_{j}$, with and without noise (light and dark blue lines, respectively), to give a visual impression of how much the added noise distorts the source data of the IRF calculations. 

### 2.6. Estimating Impulse Response Functions in the Presence of Nonstationary ARIMA Noise

The error $\varepsilon_{j}$ in Equation (4) may not only be serially correlated; it may also be nonstationary, such that its true mean changes over time. Such a random error is known as ARIMA (Autoregressive Integrated Moving Average) noise instead of ARMA noise. The standard approach to handling such cases is differencing: one subtracts the prior values from each element of the time series $x_{j}$ and $y_{j}$, yielding the “first differences” $y_{j}^{’}=y_{j}-y_{j-1}$ and $x_{j}^{’}=x_{j}-x_{j-1}$. Then Equation (4) is conventionally re-cast in terms of these first differences: (27)$$\begin{array}{c} y_{j}^{’}=y_{j}-y_{j-1}=\sum_{k=0}^{m}\beta_{k}\;x_{j-k}+\alpha +\varepsilon_{j}-\sum_{k=0}^{m}\beta_{k}\;x_{j-k-1}+\alpha +\varepsilon_{j-1}\\ =\sum_{k=0}^{m}\beta_{k}\;x_{j-k}^{’}+\alpha^{\prime }+\varepsilon_{j}^{’} \end{array},$$where the intercept $\alpha^{\prime }$ is conventionally assumed to be zero, although this will often not be strictly the case, precisely because $\varepsilon_{j}$ is nonstationary so the mean of $\varepsilon_{j}$ may not equal the mean of $\varepsilon_{j-1}$ over a finite sample from j=1…n. The conventional approach of Equation (27) leads to $\beta_{k}$ estimates that artifactually converge toward zero as k approaches m, similar to the artifact that is generated by Equation (11), as shown in Figure 1 above. The approach of Equation (27) would also greatly complicate the analysis presented in Section 3, Section 4 and Section 5 below. Therefore, a different approach will be followed here, in which first differencing is only applied to the left-hand side of Equation (4) and, similar to Equation (12) above, terms with the same lags of $x_{j-k}$ are combined. This has the effect of transforming the coefficients $\beta_{k}$ rather than the input time series x, yielding
(28)$$\begin{array}{c} y_{j}^{’}=y_{j}-y_{j-1}=\sum_{k=0}^{m+1}\beta_{k}^{’}\;x_{j-k}+\alpha^{\prime }+\varepsilon_{j}^{’} \end{array},$$where $\beta_{k}^{’}=\beta_{k}-\beta_{k-1}$, except when k=0 (whereupon $\beta_{0}^{’}=\beta_{0}$) and when k=m+1 (whereupon $\beta_{m+1}^{’}=-\beta_{m}$). These transformations can be expressed in matrix form as
(29)$$\left[\begin{array}{c} \beta_{0}^{\prime }\\ \beta_{1}^{\prime }\\ \beta_{2}^{\prime }\\ \beta_{3}^{\prime }\\ \beta_{4}^{\prime }\\ \beta_{m}^{\prime }\\ \beta_{m+1}^{\prime } \end{array}\right]=\left[\begin{array}{ccccccc} 1 & 0 & 0 & 0 & 0 & 0 & 0\\ -1 & 1 & 0 & 0 & 0 & 0 & 0\\ 0 & -1 & 1 & 0 & 0 & 0 & 0\\ 0 & 0 & -1 & 1 & 0 & 0 & 0\\ 0 & 0 & 0 & -1 & 1 & 0 & 0\\ 0 & 0 & 0 & 0 & -1 & 1 & 0\\ 0 & 0 & 0 & 0 & 0 & -1 & 1 \end{array}\right]\cdot \left[\begin{array}{c} \beta_{0}\\ \beta_{1}\\ \beta_{2}\\ \beta_{3}\\ \beta_{4}\\ \beta_{m}\\ 0 \end{array}\right]$$

One advantage of this approach is that it can use the analysis already outlined in Equations (14)–(26), with the only modifications being the use of $y_{j}^{’}$ in place of $y_{j}$ and m+1 in place of m. The results of that analysis will be in terms of $\beta_{0}^{’}\ldots \beta_{m+1}^{’}$, which can be transformed back to $\beta_{0}\ldots \beta_{m}$ by inverting Equation (29), yielding
(30)$$\left[\begin{array}{c} \beta_{0}\\ \beta_{1}\\ \beta_{2}\\ \beta_{3}\\ \beta_{4}\\ \beta_{m} \end{array}\right]=\left[\begin{array}{cccccc} 1 & 0 & 0 & 0 & 0 & 0\\ 1 & 1 & 0 & 0 & 0 & 0\\ 1 & 1 & 1 & 0 & 0 & 0\\ 1 & 1 & 1 & 1 & 0 & 0\\ 1 & 1 & 1 & 1 & 1 & 0\\ 1 & 1 & 1 & 1 & 1 & 1 \end{array}\right]\;\cdot \;\left[\begin{array}{c} \beta_{0}^{’}\\ \beta_{1}^{’}\\ \beta_{2}^{’}\\ \beta_{3}^{’}\\ \beta_{4}^{’}\\ \beta_{m}^{’} \end{array}\right].$$

One can straightforwardly combine Equations (14)–(30) to eliminate the need to use $\beta_{k}^{’}$ as an interim step between $b_{k}$ and $\beta_{k}$. The resulting procedure can be summarized as follows: first, solve Equation (14), using $y_{j}^{’}$ in place of $y_{j}$ and m+1 in place of m, to obtain the regression coefficients b and their covariance matrix $K_{bb}$. Next, transform these to the IRF coefficients β and their covariance matrix $K_{\beta \beta }$ using
(31)$$\beta =\Psi_{FD}\;\cdot \;\Psi \;\cdot \;b$$and
(32)$$K_{\beta \beta }=\Psi_{FD}\cdot \Psi \cdot K_{bb}\cdot \Psi^{T}\cdot \Psi_{FD}^{T}=\Psi_{FD}\cdot \Psi \;\cdot K_{bb}\cdot {\left(\Psi_{FD}\cdot \Psi \right)}^{T}\;,$$where $\Psi_{FD}$ is the matrix in Equation (30), which is the inverse of the first difference matrix in Equation (29). In principle, this procedure can also be straightforwardly adapted to differencing of any order κ, by differencing $y_{j}$ accordingly, using m+κ in place of m, and raising $\Psi_{FD}$ to the power κ. In practice, however, differencing beyond first order is likely to be counterproductive, because even first-differencing will magnify the high-frequency noise in the $y_{j}$ time series, and higher-order differencing will amplify this noise further. In the IRF routine, setting the parameter FD (default=FALSE) to TRUE will invoke the differencing procedure outlined in Equations (28)–(32) above. 

### 2.7. Benchmark Tests with Nonstationary ARIMA Noise

The IRF estimation approach for handling nonstationary ARIMA noise, outlined in Section 2.6 above, can be tested using benchmark tests similar to those that were used in Section 2.5 to test IRF estimates with stationary ARMA noise. As before, a synthetic random input time series $x_{j}$ is convolved with a known convolution kernel to yield the hypothetical system’s true output $y_{\mathrm{true},\;j}$. This is then corrupted by a random ARIMA noise series $y_{\mathrm{err},\;j}$ generated with R’s arima.sim function (with the order parameters set for first-order integration), which is added to $y_{\mathrm{true},\;j}$ to yield the hypothetical system’s measured output $y_{j}$. This measured output and the input series $x_{j}$ are then supplied to the IRF routine. 

In the illustrative examples shown in Figure 3, n is 10,000, m is 60, $x_{j}$ is synthetic Gaussian noise with a lag 1 serial correlation of 0.5, and the error time series $y_{\mathrm{err},\;j}$ is ARIMA (1,1,2) noise with an AR coefficient of 0.7 and MA coefficients of −0.2 and +0.2. The error time series is rescaled so that its standard deviation equals 10 times the standard deviation of $y_{\mathrm{true},\;j}$. Thus, the signal-to-noise ratio is 0.01, implying that noise dominates the measured signal. The three columns of Figure 3, similarly to Figure 2, show benchmark test results for gamma-distributed convolution kernels with shape factors of 1, 2, and 4, and the four rows of panels are interpreted similarly. The top row of panels shows that IRFs estimated from time series that are corrupted with ARIMA noise (dark blue points) correspond closely to the benchmarks (gray lines). The second row shows that the calculated standard error accurately estimates the deviations from the benchmark, the third row shows that the IRFs are unbiased, and the bottom row presents excerpts from the time series to show how much the added noise distorts the source data of the IRF calculations. 

## 3. Using Demixing Techniques to Quantify System Response to Heterogeneous Inputs

### 3.1. Demixing Multiple Impulse Response Functions

The methods outlined in Section 2 can be used to estimate impulse response functions connecting single inputs to single outputs. However, many real-world systems combine signals from multiple sources, which may themselves be correlated, and which may have overlapping effects on the output. For example, precipitation falling in different parts of a drainage basin may result in different streamflow responses (due to differences in catchment geometry or subsurface hydrological properties), which may then be lagged and dispersed differently through the channel network. As another example, one may want to infer source/receptor relationships for air pollution sources located at different distances and directions from a given receiver. In situations such as these, estimating the system’s impulse response to each individual input is both a deconvolution problem and a *demixing* problem; one needs to not only un-scramble each input’s temporally overlapping effects, but also to un-mix the different inputs’ effects from one another. 

The methods outlined in Section 1 can be straightforwardly extended to perform this combination of deconvolution and demixing. Consider the simple case of a system that combines two inputs $x_{1}$ and $x_{2}$ (which may be correlated with each other, as well as serially correlated with themselves). These two inputs are applied to corresponding fractions of the system $f_{1}$ and $f_{2}$, and are translated into the system output y over a range of lag times k=0…m by the corresponding impulse response functions $\beta_{1,k}$ and $\beta_{2,k}$:(33)$$y_{j}=f_{1}\overset{m}{\underset{k=0}{\sum }}\beta_{1,k}\;x_{1,j-k}+f_{2}\overset{m}{\underset{k=0}{\sum }}\beta_{2,k}\;x_{2,j-k}+\alpha +\varepsilon_{j},$$where the constant term α and the error term $\varepsilon_{j}$ represent any bias or error in the measured system output $y_{j}$. This simple two-input case can be straightforwardly extended to include more inputs. 

ARMA noise in the error term $\varepsilon_{j}$ can be handled analogously to Equations (5), (6) and (12)–(26) for the single-input case. Subtracting lagged copies of Equation (33), analogously to Equations (12)–(14), yields
(34)$$y_{j}=\overset{m+h}{\underset{k=0}{\sum }}b_{1,k}\;x_{1,j-k}+\overset{m+h}{\underset{k=0}{\sum }}b_{2,k}\;x_{2,j-k}+\phi_{1}\;y_{j-1}+\phi_{2}\;y_{j-2}+\ldots +\phi_{h}\;y_{j-h}+a+\xi_{j},$$where the coefficients $b_{1,k}$ and $b_{2,k}$ estimate $f_{1}\;\beta_{1,k}$ and $f_{2}\;\beta_{2,k}$, combining the individual impulse response functions $\beta_{\ell ,k}$ and the fractions $f_{\ell }$ of the system that they represent (which in principle cannot be individually determined without additional assumptions). As in Equation (14), the autoregressive coefficients ϕ also subsume the effects of moving-average noise. Equation (34) can be expressed in matrix form as (here showing the simple case of two sources, with *m* = 5, and *h* = 2): (35)$$\left[\begin{array}{c} y_{8}\\ y_{9}\\ y_{10}\\ y_{11}\\ y_{12}\\ y_{13}\\ y_{14}\\ y_{15}\\ y_{16}\\ \cdots \end{array}\right]=\left[\begin{array}{ccccccccc} x_{1,8} & \cdots & x_{1,1} & x_{2,8} & \cdots & x_{2,1} & y_{7} & y_{6} & 1\\ x_{1,9} & \cdots & x_{1,2} & x_{2,9} & \cdots & x_{2,2} & y_{8} & y_{7} & 1\\ x_{1,10} & \cdots & x_{1,3} & x_{2,10} & \cdots & x_{2,3} & y_{9} & y_{8} & 1\\ x_{1,11} & \cdots & x_{1,4} & x_{2,11} & \cdots & x_{2,4} & y_{10} & y_{9} & 1\\ x_{1,12} & \cdots & x_{1,5} & x_{2,12} & \cdots & x_{2,5} & y_{11} & y_{10} & 1\\ x_{1,13} & \cdots & x_{1,6} & x_{2,13} & \cdots & x_{2,6} & y_{12} & y_{11} & 1\\ x_{1,14} & \cdots & x_{1,7} & x_{2,14} & \cdots & x_{2,7} & y_{13} & y_{12} & 1\\ x_{1,15} & \cdots & x_{1,8} & x_{2,15} & \cdots & x_{2,8} & y_{14} & y_{13} & 1\\ x_{1,16} & \cdots & x_{1,9} & x_{2,16} & \cdots & x_{2,9} & y_{15} & y_{14} & 1\\ \vdots & \vdots & \vdots & \vdots & \vdots & \vdots & \vdots & \vdots & \vdots \end{array}\right]\cdot \left[\begin{array}{c} b_{0,1}\\ b_{1,1}\\ \vdots \\ b_{1,7}\\ b_{2,0}\\ b_{2,1}\\ \vdots \\ b_{2,7}\\ \phi_{1}\\ \phi_{2}\\ \alpha \end{array}\right]$$which can be solved by the methods outlined in Equations (17)—(26). This approach can be straightforwardly extended to any number of sources $x_{1}$, $x_{2}$, $x_{3}$, etc., although the design matrix X in Equation (35) may become rather large, with n−m rows and $n_{x}\cdot \left(m+1\right)+h+1$ columns, where n is the number of time steps, $n_{x}$ is the number of input time series, m is the maximum lag, and h is the order of AR correction. However, the IRF routine is designed to handle large problems efficiently (see Section 2.2); it can also conserve memory by creating large design matrices sequentially in chunks rather than all at once, so that the largest matrix that must be stored is only $n_{x}\cdot n$ in size. 

### 3.2. Benchmark Test

To test the approach outlined in Section 3.1 above, I generated three Gaussian random time series and convolved them with three different gamma-distributed convolution kernels, then combined the convolved signals to yield the hypothetical system’s true output $y_{\mathrm{true},\;j}$. This true output was then corrupted by a random ARMA noise $y_{\mathrm{err},\;j}$, and the error-corrupted system output $y_{j}=y_{\mathrm{true},\;j}+y_{\mathrm{err},\;j}$ was then supplied to the IRF routine, along with the three input time series. 

Two illustrative examples are shown in Figure 4: the left column was generated with three convolution kernels having roughly similar dispersion, but different lag times, whereas the right column was generated with kernels having different degrees of dispersion. In both columns, n = 10,000, m = 40, the three input signals $x_{1,j}$, $x_{2,j}$, and $x_{3,j}$ are Gaussian random noises (synthesized with a correlation of 0.8 between each pair of time series), and the error time series $y_{\mathrm{err},\;j}$ is ARMA(1,2) noise, with an AR coefficient of 0.9 and MA coefficients of −0.2 and +0.2. The error time series is rescaled so that its standard deviation equals half the standard deviation of $y_{\mathrm{true},\;j}$; thus, the signal-to-noise ratio is 4. 

The top row of panels shows that IRFs estimated from the error-corrupted time series (colored symbols) correspond closely to the benchmarks (colored lines), and that the IRFs for the three different inputs can be clearly distinguished from one another. This demixing exercise succeeds despite the clear correlations between the output signals generated from the three inputs (middle panels) and despite the distortion of the system output by ARMA noise (bottom panels). 

### 3.3. Whole-System Impulse Response Inferred from Average of Heterogeneous Inputs

The benchmark test in Figure 4 immediately raises two questions. First, in heterogeneous systems such as those shown in Figure 4, what is the whole-system impulse response (the average of the individual IRFs) to the whole-system input (the average of the individual inputs)? Second, can we correctly infer this whole-system impulse response using the methods developed in Section 2.1, Section 2.2, Section 2.3, Section 2.4, Section 2.5 and Section 2.6, if we only know the whole-system input and not the individual inputs to the different compartments of the system, with their individual impulse responses? 

From Equation (33) above, one can see that if the same instantaneous impulse were applied to the inputs $x_{1}$ and $x_{2}$ simultaneously, the expected impulse response of the system would be $f_{1}\beta_{1,k}$+$f_{2}\beta_{2,k}$. To some extent, this whole-system impulse response is inherently hypothetical, since it assumes that the inputs $x_{1}$ and $x_{2}$ are perfectly correlated, whereas estimating $\beta_{1,k}$ and $\beta_{2,k}$ in the first place requires that $x_{1}$ and $x_{2}$ are *not* perfectly correlated. Nonetheless, in many real-world cases, the individual inputs ($x_{1}$, $x_{2}$, etc.) may be sufficiently uncorrelated that their individual impulse responses ($\beta_{1,k}$, $\beta_{2,k}$ etc.) can be accurately estimated, but also sufficiently correlated that $f_{1}\beta_{1,k}$+$f_{2}\beta_{2,k}\;$will reasonably approximate how the whole system would respond, if an impulse were applied to both of the inputs simultaneously.

In many real-world cases, however, we may not know the inputs to individual components of the system, but only their average or their sum. In such cases, the true situation might be described by expressions such as Equation (33), with several inputs ($x_{1}$, $x_{2}$, etc.), but we would interpret them as if they were described by expressions such as Equation (4), with a single input x, instead. This naturally leads to the question of how the impulse response function $\beta_{k}$ that we would estimate from Equation (4) will depend on the individual IRFs and their weighting factors ($f_{1}\beta_{1,k}$, $f_{2}\beta_{2,k}$, etc.), if a system is governed by Equation (33) but is interpreted as if it is governed by Equation (4) instead. The answer can be found by applying the standard rules for first-order, second-moment propagation of covariances to the terms of the “Normal Equation” (Equation 17). Making the simplifying assumption that, although the inputs may be correlated with one another, they are not correlated with their own lags or each other’s lags, this approach yields the approximation,
(36)$$\beta_{k}=\frac{\mathrm{cov}\left(y,,,x\right)}{\mathrm{var}\left(x\right)}\approx \frac{\sum_{g,l}\frac{𝝏y}{𝝏x_{g}}\frac{𝝏x}{𝝏x_{l}}\mathrm{cov}\left(x_{g},,,x_{l}\right)}{\sum_{g,l}\frac{𝝏x}{𝝏x_{g}}\frac{𝝏x}{𝝏x_{l}}\mathrm{cov}\left(x_{g},,,x_{l}\right)}=\frac{\sum_{g,l}\beta_{g,k}f_{g}f_{l}\rho_{x_{g},x_{l}}\sigma_{x_{g}}\sigma_{x_{l}}}{\sum_{g,l}f_{g}f_{l}\rho_{x_{g},x_{l}}\sigma_{x_{g}}\sigma_{x_{l}}}$$
where $x_{g}$ and $x_{\ell }$ denote different inputs to the system, with standard deviations $\sigma_{x_{g}}$ and $\sigma_{x_{\ell }}$ and correlation $\rho_{x_{g},x_{\ell }}$, and the summations are taken over all possible pairs of g and ℓ. From Equation (36), one can see that in the simple case where all of the inputs are perfectly correlated ($\rho_{x_{g},x_{\ell }}$ = 1 for all pairs) and have the same variance (meaning that all of the inputs are identical), the impulse response function $\beta_{k}$ inferred from Equation (4) will be the average of the individual response functions $\beta_{g,k}$ weighted by the fractions $f_{g}$, and thus will equal the theoretically expected impulse response of the system derived in the previous paragraph. The inferred impulse response function $\beta_{k}$ will also be close to this theoretically expected value if the inputs are only partially correlated (0 < $\rho_{x_{g},x_{\ell }}$ < 1) or even uncorrelated with one another ($\rho_{x_{g},x_{\ell }}$ = 0), as long as they all have the same variance. Equation (36) shows that if the inputs have different variances, the inferred impulse response function $\beta_{k}$ will be approximately the average of the individual response functions $\beta_{g,k}$, weighted by $f_{g}\sigma_{x_{g}}$ if the inputs are perfectly correlated, or weighted by $f_{g}^{2}\;\sigma_{x_{g}}^{2}$ if the inputs are completely uncorrelated. If the inputs are partially correlated, the inferred impulse response function will fall between these two limiting cases. 

### 3.4. Whole-System Impulse Response Inferred from Individual Inputs

The analysis in the preceding section leads naturally to the question of whether (and under what conditions) we can correctly infer the average response of a heterogeneous system if we have only measured one of the heterogeneous inputs, but treat this time series as if it is the average input to the entire system. If we again assume that the inputs may be correlated with one another, but not with their own lags or each other’s lags (i.e., if we assume that the inputs are white noises), propagation of covariances leads to the following approximation for the whole-system impulse response that we would infer from a single input time series $x_{g}$:(37)$$\beta_{k}\approx \;\frac{\mathrm{cov}\left(y,x_{g}\right)}{\mathrm{var}\left(x_{g}\right)}\;\approx \frac{\sum_{\ell }^{}\frac{\partial y}{\partial x_{\ell }}\mathrm{cov}\left(x_{g},x_{\ell }\;\right)}{\sigma_{x_{g}}^{2}}=\frac{\sum_{\ell }^{}\beta_{\ell ,k}\;f_{\ell }\;\rho_{x_{g},x_{\ell }\;}\sigma_{x_{\ell }}}{\sigma_{x_{g}}}.$$where the summation is taken over all ℓ, including ℓ = g. In the trivial case that all of the inputs are perfectly correlated and have the same variance (i.e., they are all the same), Equation (37) predicts, unsurprisingly, that each of them will yield unbiased estimates of the whole system′s average $\beta_{k}$. If the individual inputs have the same variance but are not perfectly correlated, they will underestimate the system′s average $\beta_{k}$, with greater underestimation bias for inputs that account for small fractions of the total (small $f_{\ell =g}$) and that are weakly correlated with the other inputs (small $\rho_{x_{g},x_{\ell }}$). If the individual inputs have different variances and are substantially (but imperfectly) correlated, they can either under- or over-estimate the system′s average $\beta_{k}$, depending on how their standard deviations $\sigma_{x_{g}}$, impulse responses $\beta_{g,k}$ and input fractions $f_{g}$ compare with those of the other inputs.

### 3.5. Benchmark Test

To test the inferences derived in Section 3.3 and Section 3.4, I repeated the benchmark test shown in Figure 4, and then used the IRF routine to calculate the impulse response functions from the error-corrupted system output $y_{j}$ and the average of the three heterogeneous inputs (top panels in Figure 5), as well as each of the three inputs individually (bottom panels in Figure 5). Thus, the benchmark test in Figure 5 illustrates the IRFs that would be inferred from this heterogeneous system if only the average input were known (top panels), or if only one of the three inputs were known (bottom panels). In all cases, the inferred IRFs correspond closely to the average impulse responses of the whole system (thick gray lines in Figure 5). These results indicate that if the IRF routine is applied to systems that are heterogeneous but are not recognized as such (or not treated as such), it will yield good approximations of their average impulse responses, as long as it is supplied with the average input to the system, or with an input that is strongly correlated with that average. 

## 4. Quantifying Nonstationary Impulse Response

Many real-world systems are nonstationary, in the sense that identical inputs to the system will yield different responses at different times, due to differences in the internal state of the system or changes in external forcings. To give just a few examples from environmental science: (1) the same precipitation falling on a given landscape will typically generate a sharper runoff peak if that landscape is wet than if it is dry. (2) The runoff yield from a given volume of precipitation may also shift over much longer timescales if the landscape’s vegetation cover changes, thus altering rates of rainfall interception and evapotranspiration and thereby also altering soil moisture. (3) The coupling between atmospheric pollution sources and receptors will depend on atmospheric stability, and thus on the time of day and the prevailing weather patterns. (4) Ecosystem responses to short-term perturbations may change over time, as ambient conditions shift due to climate change. 

Systems that are stationary (in the sense that their internal processes are time-invariant) can also generate nonstationary outputs if they are subjected to nonstationary forcing, or if their outputs are contaminated by nonstationary noise. The impulse response of such systems (i.e., the coupling between their inputs and outputs) nonetheless remains stationary, and can be estimated using the methods described in Section 2 and Section 3 above. This section, by contrast, focuses on systems in which the coupling between inputs and outputs is itself nonstationary. The central question is: can the time-varying coupling between inputs and outputs in a nonstationary system be quantified from the input and output time series themselves, without direct knowledge of the system’s internal workings? 

Here, it is essential to distinguish two different cases. In the first case, the system’s impulse response changes gradually, on time scales much longer than the impulse response itself. For example, shifts in the stand density of a forest over decades or centuries may alter how the landscape responds, over timescales of hours to months, to inputs of precipitation or nutrients. Such long-term shifts in a system’s impulse response can be quantified by breaking the input and output time series into separate time intervals (in this example, perhaps separate decades) and analyzing them separately. In the methods developed in Section 2 and Section 3, this is equivalent to breaking the y vector and the X matrix in Equations (15) and (16) into horizontal blocks. This approach requires that each block is much longer than the impulse response timescale, thus minimizing the overlapping effects of inputs at the end of one block on outputs at the beginning of the next block. As long as this condition is met, this approach can be straightforwardly applied, and will not be analyzed further here. 

In the second (and decidedly less trivial) case, the system’s impulse response changes on time scales that are similar to, or shorter than, the impulse response itself. For example, a landscape’s response to precipitation will depend on its antecedent wetness, and thus on its previous precipitation inputs; in some climates, it may also depend on short-term variations in temperature and thus in the fractions of precipitation that fall as rain versus snow. In such cases, the different impulse responses of the system (to wet vs. dry conditions, for example, or snow vs. rain) are overprinted on one another. Estimating these impulse responses thus requires both deconvolution, to un-scramble the lagged effects of each input over time, and demixing, to separate the effects of the different inputs (e.g., snow vs. rain) or the effects of different system states (e.g., wet vs. dry). 

### 4.1. Deconvolving and Demixing Nonstationary Impulse Responses 

Consider the simplest case, in which there are only two types of inputs (e.g., snow vs. rain) or two states of the system (e.g., wet vs. dry) that govern the system’s impulse response. This simple case can be represented in matrix form as
(38)$$\left[\begin{array}{c} y_{6}\\ y_{7}\\ y_{8}\\ y_{9}\\ y_{10}\\ y_{11}\\ y_{12}\\ y_{13}\\ y_{14}\\ \vdots \end{array}\right]=\left[\begin{array}{ccccccccc} x_{6} & x_{5} & x_{4} & x_{3} & x_{2} & x_{1} & y_{5} & y_{4} & 1\\ x_{7} & x_{6} & x_{5} & x_{4} & x_{3} & x_{2} & y_{6} & y_{5} & 1\\ x_{8} & x_{7} & x_{6} & x_{5} & x_{4} & x_{3} & y_{7} & y_{6} & 1\\ x_{9} & x_{8} & x_{7} & x_{6} & x_{5} & x_{4} & y_{8} & y_{7} & 1\\ x_{10} & x_{19} & x_{8} & x_{7} & x_{6} & x_{5} & y_{9} & y_{8} & 1\\ x_{11} & x_{10} & x_{9} & x_{8} & x_{7} & x_{6} & y_{10} & y_{9} & 1\\ x_{12} & x_{11} & x_{10} & x_{9} & x_{8} & x_{7} & y_{11} & y_{10} & 1\\ x_{13} & x_{12} & x_{11} & x_{10} & x_{9} & x_{8} & y_{12} & y_{11} & 1\\ x_{14} & x_{13} & x_{12} & x_{11} & x_{10} & x_{9} & y_{13} & y_{12} & 1\\ \vdots & \vdots & \vdots & \vdots & \vdots & \vdots & \vdots & \vdots & \vdots \end{array}\right]\cdot \left[\begin{array}{c} b_{0}\\ b_{1}\\ b_{2}\\ b_{3}\\ b_{4}\\ b_{5}\\ \phi_{1}\\ \phi_{2}\\ \alpha \end{array}\right]$$where one type of inputs (or one set of system states) corresponding to input times 3, 4, 6, 7, 8, and 11 is shown in bold, and the other type is shown in gray. In this simple example, the maximum lag m is 3 and the AR correction order h is 2, but the principles that it illustrates are applicable to systems of any degree of complexity. The core of the problem is this: how can the impulse response functions of the black and the gray inputs be separated from one another, given that both black and gray inputs can be found in any row of Equation (38)? 

The solution to this problem is to define two different input time series, one containing the input values corresponding to the “black” time steps (and zero otherwise), and the other containing the input values corresponding to the “gray” time steps (and zero otherwise). Because one or the other of these time series is zero for every time step, their sum will exactly equal the original input time series. However, because the overlapping gray and black time series are now separated into two different variables, one can solve for their IRFs using the methods developed in Section 3 for heterogeneous systems. The resulting matrix form of the problem is
(39)$$\left[\begin{array}{c} y_{6}\\ y_{7}\\ y_{8}\\ y_{9}\\ y_{10}\\ y_{11}\\ y_{12}\\ y_{13}\\ y_{14}\\ \vdots \end{array}\right]=\left[\begin{array}{ccccccccccccccc} x_{6} & 0 & x_{4} & x_{3} & 0 & 0 & 0 & x_{5} & 0 & 0 & x_{2} & x_{1} & y_{5} & y_{4} & 1\\ x_{7} & x_{6} & 0 & x_{4} & x_{3} & 0 & 0 & 0 & x_{5} & 0 & 0 & x_{2} & y_{6} & y_{5} & 1\\ x_{8} & x_{7} & x_{6} & 0 & x_{4} & x_{3} & 0 & 0 & 0 & x_{5} & 0 & 0 & y_{7} & y_{6} & 1\\ 0 & x_{8} & x_{7} & x_{6} & 0 & x_{4} & x_{9} & 0 & 0 & 0 & x_{5} & 0 & y_{8} & y_{7} & 1\\ 0 & 0 & x_{8} & x_{7} & x_{6} & 0 & x_{10} & x_{9} & 0 & 0 & 0 & x_{5} & y_{9} & y_{8} & 1\\ x_{11} & 0 & 0 & x_{8} & x_{7} & x_{6} & 0 & x_{10} & x_{9} & 0 & 0 & 0 & y_{10} & y_{9} & 1\\ 0 & x_{11} & 0 & 0 & x_{8} & x_{7} & x_{12} & 0 & x_{10} & x_{9} & 0 & 0 & y_{11} & y_{10} & 1\\ 0 & 0 & x_{11} & 0 & 0 & x_{8} & x_{13} & x_{12} & 0 & x_{10} & x_{9} & 0 & y_{12} & y_{11} & 1\\ 0 & 0 & 0 & x_{11} & 0 & 0 & x_{14} & x_{13} & x_{12} & 0 & x_{10} & x_{9} & y_{13} & y_{12} & 1\\ \vdots & \vdots & \vdots & \vdots & \vdots & \vdots & \vdots & \vdots & \vdots & \vdots & \vdots & \vdots & \vdots & \vdots & \vdots \end{array}\right]\cdot \left[\begin{array}{c} b_{1,0}\\ b_{1,1}\\ b_{1,2}\\ b_{1,3}\\ b_{1,4}\\ b_{1,5}\\ b_{2,0}\\ b_{2,1}\\ b_{2,2}\\ b_{2,3}\\ b_{2,4}\\ b_{2,5}\\ \phi_{1}\\ \phi_{2}\\ \alpha \end{array}\right]$$which one can see is analogous to the matrix form of the heterogeneity problem shown in Equation (35), but now there is only one input time series, split between the “black” and “gray” categories, each with its own set of lag columns. Like Equation (35), Equation (39) can be solved by the methods outlined in Equations (17)–(26), and can be straightforwardly extended to any number of different types of inputs or system states, although at the potential cost of making the design matrix X rather large, depending on the number of lags m and the number of time steps n. 

### 4.2. Benchmark Tests

I conducted two benchmark tests to examine how well the deconvolution and demixing approach outlined in Section 4.1 can infer the time-varying impulse response functions of hypothetical nonstationary systems. In the first benchmark system, a random Poisson process irregularly switches among three markedly distinct impulse responses (Figure 6a), remaining with each for a random interval that averages five time steps. These impulse responses, when driven by the correspondingly colored random input time series shown in Figure 6c, generate the three components of the system output shown in Figure 6d (which would not be individually observable in real-world cases). These three components overprint one another to yield the combined system output shown by the dark gray line in Figure 6e. This, in turn, when combined with ARMA noise at a signal-to-noise ratio of 1, yields the observable system output shown by the light gray line in Figure 6e. The IRF routine, when presented with this ARMA-noise-corrupted system output and the input time series (Figure 6c), yields the estimated IRFs shown by the solid dots in Figure 6a, which generally conform to the original benchmark impulse response functions. 

The IRF routine is not provided with any information about the individual IRFs or their resulting time series shown in Figure 6d; it also has no information about the noiseless dark gray curve shown in Figure 6e, or about the nature of the added noise. Thus, this can be considered a stringent benchmark test. One could nonetheless question whether it is too simple, because it employs three benchmark IRFs that are themselves time-invariant. In the real world, nonstationary systems are likely to exhibit continuously varying impulse responses, under the influence of continuously varying drivers (e.g., temperature, solar flux, nutrient levels, etc.). 

To approximate such a case, I constructed a benchmark test in which the parameters of the benchmark impulse response vary cyclically with the hour of the day (Figure 7). Readers will notice that the system output (Figure 7g) shows a strong daily cycle. This cycle does not arise from a cycle in the input (which is purely random), but instead arises because the impulse response of the benchmark system has a stronger peak during night-time. The night-time impulse response is not larger overall—the area under each impulse response curve is the same—but it is more focused, such that more of the output comes out promptly, and less is distributed across all hours of the day. 

Figure 7a–d shows the hourly impulse responses, grouped into four six-hour periods each day. The individual hourly impulse response functions, and their six-hour averages shown by the wide colored lines in Figure 7a–d, would be unobservable in real-world cases: the only observables would be the input time series shown in Figure 7f and the noise-contaminated output time series shown by the light gray line in Figure 7g. For the benchmark test shown in Figure 7, I divided each day of the input time series into four six-hourly periods corresponding to the colors in Figure 7f (and the corresponding panels in Figure 7a–d), and solved for their impulse response functions using the methods of Section 4.1. The resulting estimates of the individual impulse response functions, as shown by the solid dots in Figure 7, are generally consistent with the benchmarks (the wide colored lines in Figure 7a–d). The deviations from the benchmark are somewhat larger in the morning hours (Figure 7b), because the impulse response is more damped and lagged, and thus the signal is relatively weaker, than for the other hours of the day.

### 4.3. Average Impulse Response of Nonstationary Systems

Many nonstationary systems may not be recognized as such, and may be analyzed as if they were stationary instead. It is worthwhile to ask what the outcomes of such analyses are likely to be. What will be the apparent stationary IRF of the system? Additionally, will this resemble, or differ from, the average of the system’s nonstationary IRFs?

The two nonstationary benchmark tests shown in Figure 6 and Figure 7 provide an opportunity to explore these questions. The solid dots in Figure 6b and Figure 7e show the ensemble IRFs that are obtained if the benchmark systems are analyzed as if they were stationary, ignoring the nonstationarity that is known to be present. In other words, the solid dots are obtained by supplying the IRF routine with a single composite input time series, rather than one that has been split into multiple inputs for the individual nonstationary categories represented by the different colors in Figure 6c and Figure 7f. Figure 6b and Figure 7e show that the resulting IRFs, shown by the solid dots, closely approximate the time-averaged benchmark impulse responses shown by the solid lines. 

Two further points are worth mentioning here. First, even if the average impulse response can be reliably quantified, it will typically be a poor predictor of the system’s time-series behavior, except on time scales that are long enough that the system’s nonstationary fluctuations average out. For example, the average IRF in Figure 7e cannot capture the daily cycles shown in Figure 7g (because they originate from the differences between the daytime and nightime IRFs), but can capture most of the variation in the daily average output. The second point worth mentioning is that, as Figure 7e shows, the time-averaged impulse response function can have a different shape from any of the individual impulse responses that make up that average. Thus, any inferred “characteristic” impulse response functions, and any inferences about their possible underlying mechanisms, should be treated with caution in systems that may exhibit nonstationarity. However, the methods outlined in this section allow such nonstationarity to be detected and quantified, even where it is obscured by overprinting of the individual impulse responses. 

## 5. Nonlinear Deconvolution

Real-world systems often exhibit nonlinearities and thresholds, in which successive additions to the input generate more-than-proportional (or, sometimes, less-than-proportional) increases in the output. In most hydrologic systems, for example, high-intensity storms generate disproportionately more runoff than low-intensity storms do. Rates of evapotranspiration, by contrast, increase roughly linearly with rates of moisture supply when moisture is limiting, but reach an upper bound when energy becomes limiting instead. Similarly, in nutrient-limited ecosystems, nutrient uptake typically increases roughly linearly with nutrient loading, but then reaches an upper limit as the ecosystem reaches nutrient saturation. This then results in nutrient exports in streams and groundwaters exhibiting threshold behavior, as nutrient loading crosses the saturation threshold.

Here it is important to distinguish two different cases. In the first case, the system’s behavior can be approximated by a succession of steady states, because the inputs vary more slowly than the impulse response of the system itself. Nonlinearities in such systems can be quantified straightforwardly by plotting their outputs as functions of their inputs. Such systems will not be considered further here.

Here, instead, I focus on the substantially less straightforward case of systems whose inputs vary on timescales shorter than their impulse responses. In such systems, nonlinear responses to input fluctuations are overprinted on one another, and thus must be deconvolved. The methods outlined in Section 2, Section 3 and Section 4 above are based on linear regression, which would seem to be an awkward tool for deconvolving input–output relationships in nonlinear systems. Nonetheless, in Section 5.1 below, I show how linear regressions can be extended to deconvolve systems’ nonlinear input–output relationships using piecewise linear approximations (as shown in Figure 8). This approach is nonparametric, in the sense that neither the shape of the impulse response function nor the form of its nonlinear dependence on the input must be specified in advance. Instead, both the impulse response function and its nonlinear dependence on the input are estimated from the input and output time series alone, even though the output combines the overlapping effects of different system responses to different magnitudes of inputs, all of which are overprinted on one another because they are convolved forward through time. Section 5.2 evaluates this approach using benchmark tests, and Section 5.3 shows that ignoring the nonlinearity in input–output relationships can lead to substantial biases in estimates of the average behavior of nonlinear systems. 

### 5.1. Deconvolving Systems’ Nonlinear Responses to Variations in Input Intensity

In principle, linear convolution relationships such as Equation (1) can be generalized by making the impulse response β dependent on not only the lag time τ, but also the lagged input intensity x(t−τ) as well: (40)$$y\left(t\right)=\int_{0}^{\infty }\beta \left(\tau ,x\left(t-\tau \right)\right)\;x\left(t-\tau \right)\;d\tau .$$

The challenge then becomes how to estimate β’s joint dependence on τ and x. In some cases, β may be separable into some function of τ (which determines the shape of the impulse response) multiplied by another function of x (which determines its magnitude). In the general case, however, β’s dependence on x can differ for different lags τ, and thus the shape of the system’s impulse response can potentially vary as a function of x (such that, for example, higher-intensity inputs could yield a larger but briefer response). This behavior should nonetheless be considered nonlinear rather than nonstationary, because β does not depend on t itself. Although β depends on x, which varies with time, a given value of x will yield the same relationship between β and τ at any time t. Thus, the convolution in Equation (40) is more appropriately termed nonlinear, rather than nonstationary, since β’s dependence on x is time-invariant, and thus the influence of x on future values of y is also time-invariant. 

The discrete counterpart to Equation (40), and thus the nonlinear counterpart to Equation (4), is: (41)$$y_{j}=\overset{m}{\underset{k=0}{\sum }}\beta_{k}\left(x_{j-k}\right)\;x_{j-k}+\alpha +\varepsilon_{j},$$where now the impulse response coefficients $\beta_{k}$ are unknown functions of the lagged input $x_{j-k}$, which in general could be different for each lag interval k. The question is how to estimate these functions. It might seem intuitively reasonable to simply divide the data set into different ranges of x, and then jointly analyze these subsets using the methods developed in Section 4 for nonstationary systems. Benchmark tests show that this approach yields biased results, because it assumes that the same intercept α applies to all ranges of x, although this is clearly not the case for linear approximations to subsets of nonlinear relationships (e.g., Figure 8). 

Instead, the approach developed here expresses the unknown dependence of $\beta_{k}$ on $x_{j-k}$ at each lag *k* using a continuous piecewise linear approximation, sometimes called a broken-stick model (see Figure 8). In Figure 8, the light blue curve shows an (unknown) continuous nonlinear function relating an input x and its lagged effects on the output y. The local slope of this light blue curve expresses how much a change in the lagged input x(t−τ) will affect the system output y(t). The light blue curve can be approximated by a continuous piecewise linear function (the dark blue dotted line) by dividing the x axis at a series of “knots” (sometimes also called “breakpoints” or “turning points”) $\kappa_{\ell }$ with $\ell =0\ldots n_{\kappa }$, as indicated by faint vertical lines and open circles in Figure 8 (these knots do not need to be evenly spaced along the x axis, although they are shown as such in the figure). 

The knots divide the range of x into $n_{\kappa }$ discrete intervals. They thus permit each value of x to be expressed in terms of a corresponding vector $x_{\ell }^{’}$ of the increments of x that lie within each interval. The $n_{\kappa }$ elements of $x_{\ell }^{’}$ express how much of each interval between knots lies at or below any given value of x:(42)$$x=\overset{n_{\kappa }}{\underset{\ell =1}{\sum }}x_{\ell }^{’},\;\begin{array}{c} x_{\ell }^{’}=\{\begin{array}{ccc} 0 & \mathrm{if} & x<\kappa_{\ell -1}\\ x-\kappa_{\ell -1}\; & \mathrm{if} & \kappa_{\ell -1}\leq x<\kappa_{\ell }\\ \kappa_{\ell }-\kappa_{\ell -1} & \mathrm{if} & x\geq \kappa_{\ell } \end{array}\\ =\max\left(0,\;\min\left(x-\kappa_{\ell -1},\;\;\kappa_{\ell }-\kappa_{\ell -1}\right)\right) \end{array}.$$

(Note that the notation for the knots κ in Equation (43) should not be confused with the covariance matrices K in Section 2.2 and Section 2.6 or the subscripts k used throughout this paper.) Equation (42) can be illustrated with a simple example. Consider a scale of x that ranges from 0 to 50 and is divided into five intervals by knots at 0, 5, 10, 20, 30, and 50. For a value of x = 27, the associated $x^{\prime }$ vector would be (5, 5, 10, 7, 0), because the entire intervals of 0–5 (5 units), 5–10 (5 units), and 10–20 (10 units) lie below x = 27, as do 7 units of the interval 20–30, but none of the interval 30–50. 

It may seem inefficient to re-express the scalar x as a vector of its increments $x_{\ell }^{’}$, but doing so facilitates the straightforward estimation of nonlinear impulse response functions using linear regression methods. As Figure 8 shows, the average impulse response coefficient $\beta_{k}\left(x\right)$ for any value of x and lag k will be closely approximated by the weighted average of the impulse response coefficients $\beta_{\ell ,k}^{’}$ associated with each increment $x_{\ell }^{’}$ of x: (43)$$\beta_{k}\left(x\right)\approx \overset{n_{\kappa }}{\underset{\ell =1}{\sum }}\beta_{\ell ,k}^{’}\;x_{\ell }^{’}/\overset{n_{\kappa }}{\underset{\ell =1}{\sum }}x_{\ell }^{’}=\overset{n_{\kappa }}{\underset{\ell =1}{\sum }}\beta_{\ell ,k}^{’}\;x_{\ell }^{’}/x.$$

This result arises because, as Figure 8 shows, the total effect on the output y of an individual input x at a lag of *k* will be approximated by the integral over the piecewise linear function shown by the dotted line, and $\beta_{k}$ is this integral divided by x. Combining Equations (41) and (43) yields a conventional multiple regression equation,
(44)$$y_{j}\approx \overset{m}{\underset{k=0}{\sum }}\left(\left(\overset{n_{\kappa }}{\underset{\ell =1}{\sum }}\beta_{\ell ,k}^{’}\;x_{\ell ,j-k}^{’}/x_{j-k}\right)\;x_{j-k}\right)+\alpha +\varepsilon_{j}=\overset{m}{\underset{k=0}{\sum }}\overset{n_{\kappa }}{\underset{\ell =1}{\sum }}\beta_{\ell ,k}^{’}\;x_{\ell ,j-k}^{’}+\alpha +\varepsilon_{j},$$which can be solved analogously to Equation (35), with a design matrix X of $n_{\kappa }\left(m+1\right)$ columns for each of the $\ell =1\ldots n_{\kappa }$ piecewise linear segments and k=0…m lags, plus an additional h+1 columns for the AR correction terms and the constant α. The resulting regression coefficients $\beta_{\ell ,k}^{’}$, when combined in weighted averages as described in Equation (43), yield the effective $\beta_{k}\left(x\right)$ for any desired value of the input x at any lag k. A vector of these $\beta_{k}\left(x\right)$ over a range of lags k defines the IRF for any given input intensity x. The corresponding uncertainties can be estimated by first-order, second-moment error propagation,
(45)$$\mathrm{SE}\left(\beta_{k}\left(x\right)\right)=\sqrt{\frac{1}{x^{2}}\;x_{\ell }^{’}\;\;{\left(K_{\beta^{\prime }\beta^{\prime }}\right)}_{k}\;x_{\ell }^{’}{}^{T}}\;,$$where ${\left(K_{\beta^{\prime }\beta^{\prime }}\right)}_{k}$ is the covariance matrix of the $\beta_{\ell ,k}^{’}$ coefficients for lag k, and $x_{\ell }^{’}$ is a row vector of the x increments. If the off-diagonal terms of this covariance matrix are ignored, Equation (45) becomes equivalent to the Gaussian error propagation formula,
(46)$$\mathrm{SE}\left(\beta_{k}\left(x\right)\right)\approx \;\sqrt{\overset{n_{\kappa }}{\underset{\ell =1}{\sum }}\mathrm{SE}{\left(\beta_{\ell ,k}^{’}\right)}^{2}\;{\left(\frac{x_{\ell }^{’}}{x}\right)}^{2}}\;.$$where SE indicates standard error. 

The impulse response of a nonlinear system can be characterized not only by how the IRF changes with lag k for a given input intensity x, but also how it changes with input intensity x for a given lag k. This can be visualized most straightforwardly by calculating the contribution to the output y from any x value at any lag k,
(47)$$y_{k}\left(x\right)=\overset{n_{\kappa }}{\underset{\ell =1}{\sum }}\beta_{\ell ,k}^{’}\;x_{\ell }^{’}=x\;\beta_{k}\left(x\right)\;,\;\mathrm{SE}\left(y_{k}\left(x\right)\right)=x\;\mathrm{SE}\left(\beta_{k}\left(x\right)\right)$$

Plotting Equation (47) over a range of x values will visually reveal the nonlinearity of y’s response to x at a given lag k. Since this plot will be piecewise linear between each pair of knots, everything that is known about $y_{k}\left(x\right)$ can be summarized by evaluating Equation (47) for each combination of knots$\;x=\kappa_{\ell }$ and lags k. Conversely, $\beta_{k}\left(x\right)=y_{k}\left(x\right)/x\;$will in general not be piecewise linear as a function of x, because x appears in the denominator of Equation (43). 

The calculations outlined above are implemented in the nonlinIRF function within the R script IRFnnhs.R. The question remains of how the knots $\kappa_{\ell }$ should be selected. In the R script provided here, the task of choosing these knots is left to the user. Users should balance the twin objectives of keeping the segments between knots short enough that they will approximate the (unknown) nonlinear response curve, while simultaneously not making them so short that they contain too few x values to allow the individual $\beta_{\ell ,k}^{’}$ to be accurately estimated. In principle, one could also use iterative search algorithms to find optimal sets of knots, using criteria such as minimization of expected prediction error, but such approaches have not been implemented in the nonlinIRF function. 

### 5.2. Benchmark Tests

I conducted several benchmark tests to examine how well the approach outlined in Section 5.1 captures nonlinear impulse response functions. All of these tests start with a 10,000-point log-normally distributed white-noise random input time series x (e.g., Figure 9a), and convolve each point of this time series with gamma distributions whose parameters vary as nonlinear functions of x. In the example shown in Figure 9b, each point is convolved with a gamma distribution with a shape factor of 2 and a mean lifetime of 10, whose amplitude scales as $x^{2}$, with the result that the contribution of each point to the output y scales as $x^{3}$. Diverse nonlinear functions of x are used to re-scale the amplitude of the gamma-distributed convolution kernel (Figure 10 and Figure 11), or to re-scale its shape factor or mean lifetime (Figure 12 and Figure 13). The overprinted nonlinear effects of all of the x inputs (e.g., Figure 9b) comprise the single output time series y, which is then corrupted by ARMA(1,2) noise with an AR coefficient of 0.9 and MA coefficients of −0.2 and +0.2, with the noise variance adjusted to achieve a signal-to-noise ratio (SNR) of 4. A sample of the noise-corrupted system output is shown by the light gray line in Figure 9c; one can see that the noise has substantially obscured many of the smaller-amplitude features in the true system output (shown by the dark gray line). 

The input time series x and the noise-corrupted output time series y, along with a set of knots κ, are supplied to the nonlinIRF routine, which applies the approach outlined in Equations (42)–(47) above and calls the IRF routine to perform the core calculations. Note that these algorithms are not provided with any information about the benchmark impulse responses or their nonlinear dependence on x, beyond what is contained in the time series themselves, where the effects of the individual inputs are overprinted on one another. 

Figure 10 shows example results for a simple case in which the amplitude of the impulse response function scales proportionally to the input x (Figure 10a,b) and thus the system response $y_{k}\left(x\right)$ scales proportionally to $x^{2}$ (Figure 10c,d). The estimated impulse response functions and system responses $y_{k}\left(x\right)$ (dots) generally agree with the benchmarks (lines, evaluated at the knots $x=\kappa_{\ell }$), with the exception of the weakest responses (red dots, corresponding to the smallest values of x), which often deviate from their benchmarks by more than their standard errors. The weak signals from these small inputs are swamped by the much larger noise (Figure 9b,c), and distorted by coincidental correlations with the much stronger signals from much larger inputs. These deviations are much less visible in plots of the system response $y_{k}\left(x\right)$, almost vanishing into the x axis (e.g., Figure 10c); they are only evident in plots of the impulse response function $\mathrm{IRF}=\beta_{k}\left(x\right)=y_{k}\left(x\right)/x$ (e.g., Figure 10a). Thus, although the impulse response per unit input (the IRF) can exhibit substantial deviations when both the input and the resulting impulse response are small (because the system output y is relatively insensitive to small values of x), for the same reason, these deviations will have only a minimal effect on the overall system behavior. By contrast, the stronger signals that correspond to higher input values are more reliably quantified in both the IRF and the system response $y_{k}\left(x\right)$, although with relatively large error bars at the upper tail of the x distribution (blue dots) due to the relative scarcity of data points near that upper tail. 

As Figure 10d shows, the nonlinearity in the benchmark system is well captured by relationship between the input x and the peak system response $y_{k}\left(x\right)$. Looking beyond this simple quadratic relationship, Figure 11 shows that the nonlinIRF routine can reliably estimate a wide range of nonlinear relationships between the system input and output (even though, as mentioned above, it has no prior information about either the shape of the system’s impulse response or its nonlinear dependence on the input). 

In the benchmark tests shown in Figure 10 and Figure 11, the benchmark impulse response function always has the same shape (a gamma distribution with a shape factor of 2 and a mean lifetime of 10), and is simply re-scaled as a function of the input x. However, what if changes in the input x do not merely re-scale the impulse response, but change its shape? In Figure 12, I show the results of a benchmark test in which the impulse response function is still a gamma distribution with a shape factor of 2, but its mean lifetime decreases from about 40 days for the smallest inputs to about 5 days for the largest inputs. Figure 13 shows a further benchmark test in which the shape factor of the gamma distribution decreases from roughly 3 for small inputs to roughly 0.7 for large inputs. In both of these cases, the IRFs and system responses $y_{k}\left(x\right)$ closely match their benchmarks. Considered together, Figure 10, Figure 11, Figure 12 and Figure 13 show that the approach outlined in Section 5.1 can accurately infer nonlinear input–output relationships, even when those relationships are convolved over time and are partly obscured by ARMA noise. 

The nonlinearities considered here bear a superficial resemblance to Hammerstein systems, for which several identification algorithms have been proposed (e.g., [22,23]), but there are two important differences. First, Hammerstein systems consist of two parts connected in series: the input is first transformed by a nonlinear function, which in turn drives a linear AR or ARMA system. Thus, the entire system response scales nonlinearly as a function of the input. In systems such as Equations (40) and (41), by contrast, the system response may vary in shape, not just in scale, as the input changes. Second, in Hammerstein systems, the autoregressive or ARMA behavior is assumed to originate within the system itself, whereas here it is assumed to characterize the noise. Thus, the primary focus here is on estimating the impulse response coefficients $\beta_{k}\left(x\right)$ and their nonlinear dependence on the input x, rather than the AR coefficients ϕ. This has important implications for the solution method, as described at the end of Section 2.2 above.

It bears emphasis that the nonlinearities that are addressed by this approach are nonlinear relationships between the input x and the output y. Such relationships should be handled using the methods presented here, rather than the demixing methods presented in Section 4. Impulse response functions may also be nonlinear functions of other variables besides x, but those are more properly cases of nonstationarity that should be handled by the methods of Section 4, rather than those presented in Section 5.1 above. 

### 5.3. Biases in the Apparent Average Impulse Response of Nonlinear Systems

Conventional methods for estimating impulse response functions are based on the premise that the underlying system exhibits linearity and stationarity, that is, that the function β(τ) in Equation (1) and the coefficients $\beta_{k}$ in Equation (3) depend only on the lag time τ or the lag index k, and are otherwise constant (i.e., stationary), and in particular are independent of the input x (i.e., linear). What if the real-world system is not linear and stationary, but nevertheless is analyzed as if it were? The benchmark tests in Section 4.3 above examined what would happen if time series from nonstationary (but linear) systems were analyzed as if those systems were stationary instead. The results showed that such cases can be expected to yield estimates (dots in Figure 6b and Figure 7e) that approximate the average of the actual, nonstationary impulse response (lines in Figure 6b and Figure 7e). 

This is generally true for nonlinear systems. Insight into the reasons why can be gained from a simple theoretical analysis. Consider the following relationship,
(48)$$y_{j}=\beta_{j}\;x_{j}+\alpha +\varepsilon_{j},$$which can be viewed as an analogue to Equation (3) in which the lags *k* are ignored (a simplification) and the constant β has been replaced by the time-varying $\beta_{j}$ (a complication). Because $\beta_{j}$ is not constant, and potentially could be different for every time step j, Equation (48) is not a regression equation. However, what if we did not know that? What if our data originated from a system described by Equation (48), but we analyzed it as if it came from the regression equation $y_{j}=\beta \;x_{j}+\alpha +\varepsilon_{j}$ instead? How will a regression estimate of the single coefficient β (here denoted $\hat{\beta }$ to distinguish it from the true value β) depend on the (unknown) values of the individual $\beta_{j}$, including their possible dependence on the input values $x_{j}$? 

The answer to this question, derived without approximations in Appendix A of [20], is
(49)$$\hat{\beta }-\bar{\beta }=\bar{x}\;\frac{\mathrm{cov}\left(\beta_{j},\;x_{j}\right)}{\mathrm{var}\left(x_{j}\right)}+\frac{\langle (\beta_{j}-\bar{\beta })\rangle \;{\langle (x_{j}-\bar{x})\rangle }^{2}}{{\left(x_{j}-\bar{x}\right)}^{2}}+\frac{\mathrm{cov}\left(\varepsilon_{j},\;x_{j}\right)}{\mathrm{var}\left(x_{j}\right)},$$where overbars and angled brackets indicate averages. Equation (49) cannot be evaluated in practice because the individual β coefficients are unknown, but it nonetheless demonstrates how their properties influence the regression estimate $\hat{\beta }$. Equation (49) says that this regression estimate will equal the true mean $\bar{\beta }$ of the individual β coefficients, plus the three terms on the right-hand side. The first of these terms is the average value of x, multiplied by the regression slope of the relationship between x and β. This first term shows that if β is positively correlated with x, the regression estimate $\hat{\beta }$ will be inflated relative to the true mean $\bar{\beta }$. The second term of Equation (49) is a weighted average of the deviations of the β coefficients from their mean, where the weights are the leverages of the individual values of x (their squared deviations from their mean). This second term shows that $\hat{\beta }$ will be biased upward if the relationship between β and x is upward-curving, and downward if the relationship is downward-curving. The third term of Equation (49) says that $\hat{\beta }$ could also be biased by correlations between the errors ε and the inputs x, as can arise in “hidden variable” problems, although these correlations should be zero (within statistical noise) if the errors ε are truly random. 

In summary, Equation (49) shows, consistent with the results reported in Section 4.3 above, that input and output time series from nonstationary systems will generally yield unbiased estimates $\hat{\beta }$ of their average impulse response $\bar{\beta }$, even if their nonstationary character is overlooked in the analysis, as long as these systems are not also nonlinear (that is, as long as their impulse responses β are independent of the input x). Conversely, however, Equation (49) also shows that input and output time series from nonlinear systems will generally yield biased estimates $\hat{\beta }$ of their average impulse response $\bar{\beta }$, unless their nonlinearity is taken into account. 

Figure 14 illustrates this bias for four different nonlinear benchmark systems from Figure 10, Figure 11, Figure 12 and Figure 13. The gray lines depict the impulse responses $\beta_{k}$ for each lag k at the knot values $x=\kappa_{\ell }$ of the inputs, and the blue lines depict the average impulse response functions ${\bar{\beta }}_{k}$, averaged over all input values. The red points show naïve estimates of the average impulse response functions, obtained using the methods of Section 2.1, Section 2.2, Section 2.3 and Section 2.4, without accounting for the systems’ nonlinear behavior. If $y_{k}\left(x\right)\sim x^{2}$ and thus $\beta_{k}\sim x$ (Figure 14a and Figure 10), these naïve estimates exaggerate the true average impulse response by nearly a factor of 3, because β is positively correlated with x and also with the leverage of x, and thus the first two terms of Equation (49) are both positive. By contrast, in a system governed by Michaelis–Menten saturation, in which $y_{k}\left(x\right)\sim x/\left(1+x\right)$ and thus $\beta_{k}\sim 1/\left(1+x\right),$ as shown in Figure 14b and Figure 11e, the naïve estimate is roughly one-third of the true average impulse response, because β is negatively correlated with x and with the leverage of x, and thus the first two terms of Equation (49) are both negative. In benchmark systems in which different input values yield impulse responses with different mean response times (Figure 14c and Figure 12) or shape factors (Figure 14d and Figure 13), naïve estimates can differ markedly from the shapes of the true average impulse response. 

The substantial biases in these naïve estimates motivate the question of how the average impulse response functions ${\bar{\beta }}_{k}$ can be estimated in nonlinear systems. The straightforward answer is that once the incremental impulse response coefficients $\beta_{\ell ,k}^{’}$ have been estimated by regression via Equation (44), one can use Equations (43) and (47) to calculate $\beta_{k}\left(x\right)$ and $y_{k}\left(x\right)$ for any value of x, and thus calculate their average from the distribution of x. These estimates of ${\bar{\beta }}_{k}$, shown by the blue dots in Figure 14, generally follow the average benchmarks shown by the blue lines. 

## 6. Capturing Multiscale Impulse Response with Unevenly Spaced Piecewise Linear IRFs

Many systems exhibit impulse response over multiple time scales, often starting with a sharp, brief initial response, which then transitions to a persistent lower-level response that decays much more slowly. It is difficult to capture these different timescales with a conventional IRF defined by coefficients for evenly spaced lag times. 

In principle, of course, one could simply use the methods of Section 2, Section 3, Section 4 and Section 5 to evaluate IRF coefficients over hundreds or thousands of lags, extending out to the longest lag time of interest. Doing so, however, is computationally inefficient: although it is possible to evaluate IRFs over thousands of lags, the necessary matrix operations become time-consuming (see Section 2.2), although modern matrix solvers make this issue much less pressing than it once was. A more serious problem is that it is also *statistically* inefficient to estimate too many IRF coefficients from inherently limited data, because the finite information those data contain must be spread among the many coefficients to be estimated. This increases the uncertainties in the estimated IRF at each lag, making it difficult to accurately estimate the slowly varying long tails of many real-world impulse response functions.

The long tails could be better constrained if the input and output time series were aggregated to coarser time steps, with a corresponding reduction in the number of IRF coefficients that must be estimated. The obvious drawback of this approach is that one loses the ability to accurately quantify the system’s short-term impulse response.

Another common approach to this problem is to assume that the IRF has a known functional form (e.g., a gamma distribution or a set of exponentials), and to estimate the parameters for that function. However, this typically results in a model that is nonlinear in its parameters, with the implication that parameters must be estimated by computationally intensive iterative search methods (e.g., [24]) and one can never be certain that the single best set of parameters has been found (the local optimum problem). Such approaches also depend critically on the assumption that the chosen function is the correct one, which is usually impossible to verify.

### 6.1. Piecewise Linear Approximations to Impulse Response Functions

In principle, IRFs are continuous functions of lag time, so they can potentially be approximated by piecewise linear functions, with the vertices connecting the linear segments being closely spaced at short lag times (where the IRF is changing rapidly), and more widely spaced at longer lag times (where the IRF is only changing slowly). Such a piecewise linear IRF is completely determined by the values at the vertices between the linear segments. Thus, it is only necessary to constrain the relatively few (unevenly spaced) coefficients for the vertices, rather than the conventional IRF coefficients for every lag.

Here I demonstrate how this can be efficiently solved as a linear regression problem. As shown in Figure 15, the nonlinear IRF shown by the light blue curve would conventionally be evaluated at the evenly spaced light blue dots. However, it can also be approximated by a piecewise linear function (the dark blue dashed lines) between specified knots $\kappa_{\ell },\;\ell =1\ldots n_{\kappa }$, shown by the open circles. In this piecewise linear approximation, each of the IRF coefficients $\beta_{k}$ in Equation (3) can be expressed instead in terms of the IRF coefficients of the two adjacent knots $\beta_{\kappa }^{*}$, as follows (here showing an example where $\beta_{k}$ lies between the third and fourth knots):(50)$$\beta_{k}=\beta_{3}^{*}\left(\frac{\kappa_{4}-k}{\kappa_{4}-\kappa_{3}}\right)+\beta_{4}^{*}\left(\frac{k-\kappa_{3}}{\kappa_{4}-\kappa_{3}}\right)$$Substituting each of these $\beta_{k}$ into Equation (3) and rearranging terms yields
(51)$$y_{j}=\overset{n_{\kappa }}{\underset{\ell =1}{\sum }}\beta_{\ell }^{*}\left[\overset{\left(\kappa_{\ell }\right)-1}{\underset{k=\kappa_{\ell -1}}{\sum }}\left(\frac{k-\kappa_{\ell -1}}{\kappa_{\ell }-\kappa_{\ell -1}}\right)x_{j-k}+\overset{\kappa_{\ell +1}}{\underset{k=\kappa_{\ell }}{\sum }}\left(\frac{\kappa_{\ell +1}-k}{\kappa_{\ell +1}-\kappa_{\ell }}\right)x_{j-k}\right],$$where the terms in curved brackets define a set of triangular weighting functions over lag time surrounding each knot lag $\kappa_{\ell }$, as shown below the x-axis in Figure 15. This approach converts Equation (4) into an equivalent regression with transformed variables,
(52)$$y_{j}=\overset{n_{\kappa }}{\underset{\ell =1}{\sum }}\beta_{\ell }^{*}\;x_{j,\ell }^{*}+\alpha +\varepsilon_{j},$$where the $x_{j,\ell }^{*}$ are defined by the quantity in square brackets in Equation (52). Equation (52) can then be solved by the methods outlined in Section 2, with the proviso that the corrections for ARMA and ARIMA noise will be less exact because the $x_{j,\ell }^{*}$ are not evenly spaced in lag time.

Readers may note a superficial similarity between Figure 15 and Figure 8. There is an important difference, however: in Figure 8, each IRF coefficient $\beta_{k}$ is expressed as a piecewise linear function of the input intensity x, whereas in Figure 15, the IRF coefficients are expressed as piecewise linear functions of the lag k. Whereas the broken-stick model in Figure 8 multiplies the number of coefficients that must be estimated (replacing each of the m+1 IRF coefficients with $n_{\kappa }$ knots, for a total of $\left(m+1\right)*n_{\kappa }$ knots in total), the broken-stick model in Figure 15 reduces the number of coefficients that must be estimated, substituting $n_{\kappa }$ knots for all m+1 IRF coefficients.

The approach outlined above is invoked in the IRFnnhs.R script by setting the nk parameter to an integer between 3 and m, instead of the default value of nk = 0. The script then estimates the piecewise linear IRF connecting nk knots, placed at lags 0, 1, and a geometric progression of lags between 1 and m (or as close to a geometric progression as possible, given that the lags are integers). 

### 6.2. Benchmark Test

I tested the approach outlined above using a benchmark convolution kernel that combines a sharp peak at short lags and a weaker but more persistent long-term response (as shown in the gray curves in Figure 16). This known convolution kernel was convolved with a 10,000-point random input time series to generate the benchmark system’s true output, which was then corrupted by noise at a signal-to-noise ratio of 1. I then deconvolved the noise-corrupted time series using the IRF methods developed in Section 2 (left column, Figure 16) and the piecewise linear approach of Section 6.1 (middle and right columns, Figure 16, showing 50 and 20 knots, respectively). The three rows of Figure 16 show results for white noise (top row), moderately autocorrelated AR(1) noise with an AR coefficient of ρ=0.5 (middle row), and more strongly autocorrelated AR(1) noise with ρ=0.95 (bottom row). 

One can see from Figure 16 that the sharp short-term response is accurately depicted in all cases. However, the weaker, longer-term response is either obscured, distorted, or accurately revealed, depending on both the method that is used and the autocorrelation in the added noise. When the noise is uncorrelated (top row) or moderately autocorrelated (middle row), the longer-term response is obscured by scatter in the regular, evenly spaced IRF (Figure 16a,d). The piecewise linear IRF, by contrast, can clearly distinguish the longer-term response from zero (indicated by the dashed gray line in Figure 16b,c,e,f), because it averages over the random fluctuations in the noise. This averaging becomes somewhat less effective as the noise becomes more strongly autocorrelated, as shown by the progression from the top row to the bottom row in Figure 16. When the first-order autocorrelation in the noise is ρ=0.95, implying a characteristic decorrelation time of roughly 20 time steps (bottom row of Figure 16), the scatter in the regular IRF is greatly reduced but replaced with artifactual coherent distortions (Figure 16g). These distortions are somewhat reduced, but still persist, in the piecewise linear IRF with 50 knots (Figure 16h), but are more effectively removed by the piecewise linear IRF with 20 knots (Figure 16i). This is because when the knots are more widely spaced, their averaging timescales become larger in comparison to the decorrelation time of the noise, and thus the noise can be more effectively averaged out.

## 7. Discussion

The benchmark tests presented in Section 2.5, Section 2.7, Section 3.2, Section 3.5, Section 4.2, Section 4.3, Section 5.2, Section 5.3, and Section 6.2 demonstrate that the methods developed here can accurately estimate the impulse response of heterogeneous, nonstationary, and nonlinear systems, even in the presence of autoregressive and nonstationary noise. These tests are not comprehensive, however, because real-world applications will entail many diverse systems with different impulse response characteristics, forcing time series, and potentially contaminating noises. No feasible benchmark testing program could test all of these possibilities, in all possible combinations. Thus, users are encouraged to benchmark these methods using synthetic data reflecting their particular applications.

Section 2, Section 3, Section 4, Section 5 and Section 6 each present solutions to specific problems that arise in estimating impulse response functions from real-world data: system heterogeneity, nonstationarity, and nonlinearity, autoregressive and nonstationary noise, and multiscale impulse response. Real-world cases will often combine two or more of these problems. Thus, we might need to estimate the multiscale impulse response (Section 6) of a system that may also be nonlinear (Section 5), nonstationary (Section 4), and/or heterogeneous (Section 3), and that may also generate signals that are obscured by autoregressive or nonstationary noise (Section 2). The IRFnnhs.R script can handle any combination of such problems, subject of course to any limitations in the information content of the available input and output time series. However, benchmark testing all of the possible permutations is not feasible, so users will need to conduct appropriate benchmark tests for the combinations of problems that they face in individual real-world problems.

Although the methods developed here can accurately estimate the impulse response of many different systems (including, in particular, systems that are heterogeneous, nonstationary, or nonlinear), three potentially important limitations should be noted. First, one should not expect the resulting impulse response functions to accurately predict these systems’ time-series outputs, unless their behavior is completely described by their IRFs, which will often not be the case. The approaches developed here are primarily intended to gain insight into how systems work, using IRFs as a measure of their integrated behavior. In this context, the key question is whether these methods can accurately capture systems’ nonlinear, nonstationary, and heterogeneous impulse responses (which they indeed can, according to the benchmark tests presented here), not whether those IRFs, by themselves, accurately predict system outputs over time. 

Second, although the methods developed here can yield accurate IRF estimates despite substantial autocorrelated or even nonstationary noise in the system output, they do not correct for errors in the system inputs (often termed the “errors-in-variables” problem). Techniques for handling the errors-in-variables problem in this context are currently under development and may be presented in a future paper. 

Third, the term “nonlinear” as used here refers to systems whose impulse response functions depend nonlinearly on the input intensity, and thus can be quantified by nonlinear deconvolution techniques such as those described in Section 5. Similar terminology is often used to refer to a broad class of nonlinear dynamical systems—that is, systems of nonlinear differential equations characterized by bifurcations and chaotic dynamics. Such systems are not described by linear convolutions such as Equation (1), or nonlinear convolutions such as Equations (40) and (41), and thus cannot be deconvolved by the methods outlined here. Simply put, such systems do not have impulse response functions, so the methods outlined here cannot estimate them.

Readers should keep in mind that the reliability of any deconvolution method, including the methods described here, will depend on the autocorrelation behavior (and thus the frequency content) of the input to the system. In convolutional systems such as Equation (1), the impulse response will be poorly constrained at any frequency for which the input to the system exhibits little or no variation. This is inherent in the mathematics of convolution, and is independent of the particular deconvolution methods that are used. The principles can be easily seen by re-casting the linear convolution in Equation (1) as its Fourier transform:(53)Y(f)=B(f)·X(f)where Y(f), B(f), and X(f) are the (complex) Fourier transforms of y(t), β(t−τ), and x(t−τ). From Equation (53), one can see that Y(f) will have no spectral power at any frequency f where X(f) contains no spectral power. Thus, the deconvolution estimate of B(f),
(54)B(f)=Y(f)/X(f)will be undefined at that frequency, because Equation (54) will be dividing zero by zero. For nonlinear convolutions the situation is more complex, because an input X(f) at any given frequency f will yield outputs not only at f but also at its harmonics, and thus Equations (53) and (54) will no longer apply. 

Nonetheless, the principle remains that broadband inputs will generally yield more reliable estimates of the impulse response function than narrowband inputs will. In most of the benchmark tests shown here, the inputs are Gaussian white noise (and hence are ideal broadband signals), except for Figure 2, Figure 3, and Figure 16, in which the inputs are Gaussian noise with a lag 1 correlation of 0.5. Fortunately, many real-world systems also have broadband inputs, and thus may be well suited to deconvolution approaches such as those outlined here. Users with any concerns in this regard are encouraged to explore them with benchmark tests tailored to the characteristics of their own systems and data sets. 

Given the recent widespread interest in machine learning models, it may be helpful to contrast the present approach with artificial intelligence approaches. Machine learning methods typically attempt to predict the system output as accurately as possible, using flexible, parameter-rich models calibrated against large sets of training data. Whereas the functional relationships within machine learning models are often difficult or impossible to visualize, the focus of the present approach is precisely to reveal the functional relationships between systems’ inputs and outputs—their impulse response functions—and make them visible to the user. These impulse response functions would not be promising candidates for estimation by machine learning models, because the true impulse response functions are unknown, and thus no training data exist to estimate them. Whereas machine learning approaches are unguided explorations of all possible relationships between inputs and outputs, the present approach is strongly guided by the *a priori* assumption that the dominant relationships (or at least the relationships of primary interest) between the inputs and outputs are impulse response functions. The present approach is designed to encourage users to query their data iteratively, testing alternative explanations for the inferred input–output relationships. In this respect, the present approach primarily focuses on human learning rather than machine learning. 

The analyses presented here suggest a potentially promising extension. If the impulse response function $\beta_{k}$ can be estimated accurately enough, it is in principle straightforward to deconvolve the system output time series $y_{j}$ by $\beta_{k}$ to estimate the input time series $x_{j}$. If, in addition, the “errors in variables” problem can be successfully handled, such that the input errors average out in estimating $\beta_{k}$, this deconvolution approach could potentially even estimate the input time series $x_{j}$ more accurately than it can be measured (given sufficiently accurate measurements of the output time series $y_{j}$). Initial tests of this approach are promising, but will need to be refined in future work. 

## 8. Summary and Conclusions

Systems’ impulse responses can be useful measures of their integrated behavior. However, real-world systems are often heterogeneous, nonstationary, and nonlinear, and their time series often have autoregressive or even nonstationary errors. All of these characteristics are problematic for conventional methods for estimating impulse response functions. Here, I have presented a suite of data-driven, model-independent, nonparametric methods for inferring the impulse responses of systems that may be heterogeneous, nonstationary, or nonlinear, and whose time series may be substantially corrupted by poorly-behaved noise. The IRFnnhs.R script can solve these problems efficiently, even including problems that generate design matrices with hundreds or thousands of columns and millions of rows. 

These methods can estimate impulse response functions from time series that are substantially contaminated by ARMA noise (Section 2.2 and Section 2.3, Equations (12)–(26)) and nonstationary ARIMA noise (Section 2.6, Equations (27)–(32)) through a single inversion of a linear system of equations, without requiring an iterative search of the parameter space. Benchmark tests demonstrate that these methods can effectively handle large amounts of ARMA noise (Figure 2) and ARIMA noise (Figure 3), and that they are orders of magnitude faster than, for example, R’s arima function, which searches iteratively. 

These methods can be straightforwardly extended to not only un-scramble (deconvolve) the lagged effects of inputs over time, but also to separate (demix) the overlapping effects of inputs to individual compartments of heterogeneous systems (Section 3.1, Equations (33)–(35)). Benchmark tests demonstrate that this approach can effectively distinguish even broadly similar impulse responses from one another, even when they are overprinted on one another in the system output (Figure 4). If a system’s heterogeneities are ignored, and one instead deconvolves the combined output by the combined input, the methods of Section 2 will generally yield a good approximation to the system’s average impulse response—which may differ greatly from the impulse responses of its individual compartments (Figure 5). 

This deconvolution–demixing approach can be further extended to quantify the IRFs of systems that are nonstationary on timescales shorter than their impulse responses themselves, such that their different impulse responses are overprinted on each other (Section 4.1, Equation (39)). Even though these individual impulse responses are obscured in the system output, benchmark tests show that they can be accurately detected and quantified (Figure 6 and Figure 7). In systems that are nonstationary but are not recognized as such, the methods of Section 2 will generally yield a good approximation to the time-averaged impulse response (Figure 6b and Figure 7e). 

The deconvolution–demixing approach can also be extended to estimate impulse response functions that depend nonlinearly—in shape or amplitude or both—on the system input. A broken-stick model of the impulse response at each lag (Figure 8) allows nonlinear impulse responses to be efficiently quantified using purely linear algebra (Equations (42)–(47)) without making any *a priori* assumptions about either the shape of the impulse response function or its nonlinear input-dependence. Benchmark tests demonstrate that this approach yields realistic estimates of both the impulse response function and its nonlinear variability, even in systems whose inputs vary much more rapidly than their impulse responses—and thus whose impulse responses are overprinted on one another in the output (Figure 9, Figure 10, Figure 11, Figure 12 and Figure 13). If nonlinear systems are not recognized as such, and instead are analyzed as if they were linear, their apparent impulse response functions can deviate substantially, in both shape and amplitude, from their true average impulse response (Figure 14). These deviations can be explained by a relatively straightforward analysis of statistical moments (Equations (48) and (49)). 

Many systems are characterized by a combination of strong short-term impulse response and much weaker, but more persistent, long-term response. IRFs that extend to long lags will often be too noisy to reveal this long-term response, because their many coefficients will dilute the information content of the input time series. In such cases, a piecewise linear approximation to the IRF (Equations (50)–(52)) can be used to evaluate the IRF at knots that are unevenly spaced in lag time (Figure 15), allowing the system’s long-term response to be more accurately quantified (Figure 16) while also accurately portraying its much sharper short-term response.

Finally, it is worth noting that the benchmark tests performed here included significant levels of (often poorly behaved) noise, and the benchmarks did not necessarily conform to the assumptions underlying the analysis methods. For example, the nonlinear analysis in Section 5 assumed that the impulse response at each lag is a piecewise linear function of the input, but the benchmarks used to test those methods were not. Likewise, the analysis algorithms had no information about how the benchmarks were generated, beyond whatever they could infer from the input and output time series (the latter being substantially contaminated with badly behaved noise). Thus, these benchmark tests are more rigorous than many found in the literature, in which the benchmarks and analysis methods are tailored to fit one another.

## Figures and Tables

**Figure 1 sensors-22-03291-f001:**
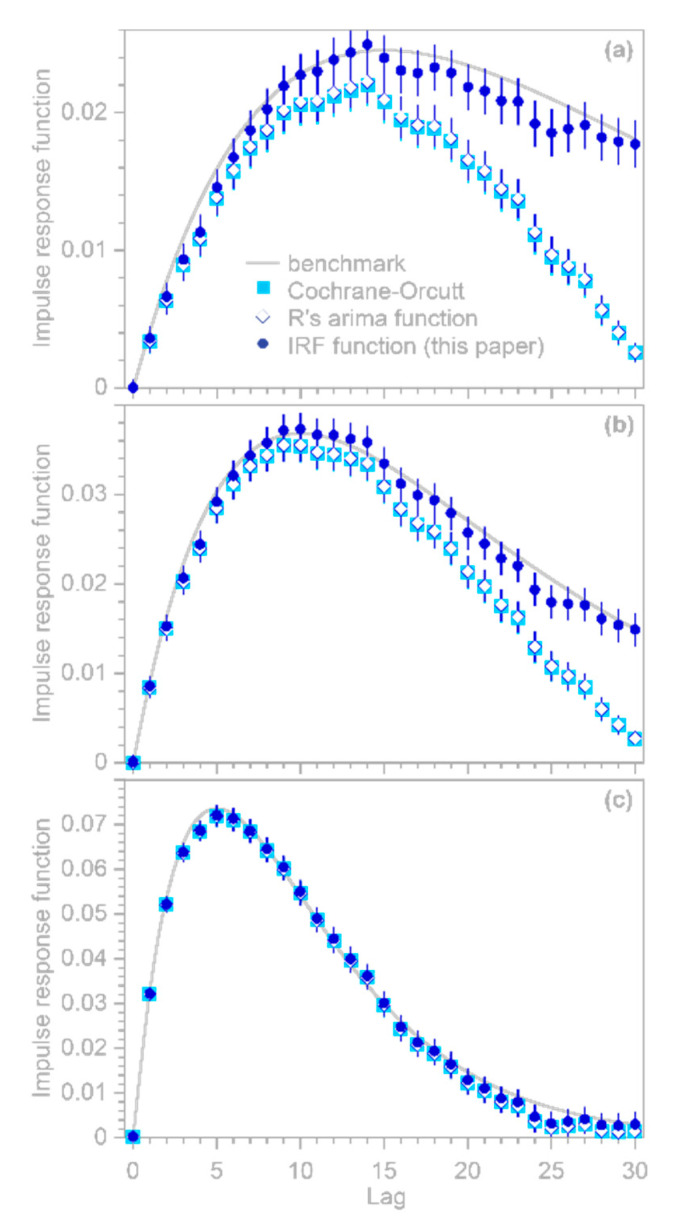
Impulse response functions estimated by the Cochrane–Orcutt procedure (Equations (9)–(11), light blue squares), R’s arima function (open diamonds), and the IRF function presented here (Equations (12)–(26), dark blue circles). Gray lines show benchmark convolution kernels (gamma distributions with shape factor α = 2 and means τ of 30, 20, and 10 lag units in panels (**a**), (**b**), and (**c**), respectively), which were convolved with a Gaussian white noise time series of length n = 2000 to simulate the system output. First-order autoregressive noise with ρ = 0.9 was added to the system output at a signal-to-noise ratio of 4. The autoregressive coefficient of ρ = 0.9 was supplied as a known parameter to the Cochrane–Orcutt procedure and R’s arima function. Impulse response functions estimated by the Cochrane–Orcutt procedure and R’s arima function (light blue squares and open diamonds, respectively) converge to nearly zero within the range of analyzed lags, even if the true convolution kernel does not (panels (**a**,**b**)). If the true convolution kernel converges to nearly zero within the range of analyzed lags (which will not be known in practice), all three methods yield nearly identical results (panel (**c**)).

**Figure 2 sensors-22-03291-f002:**
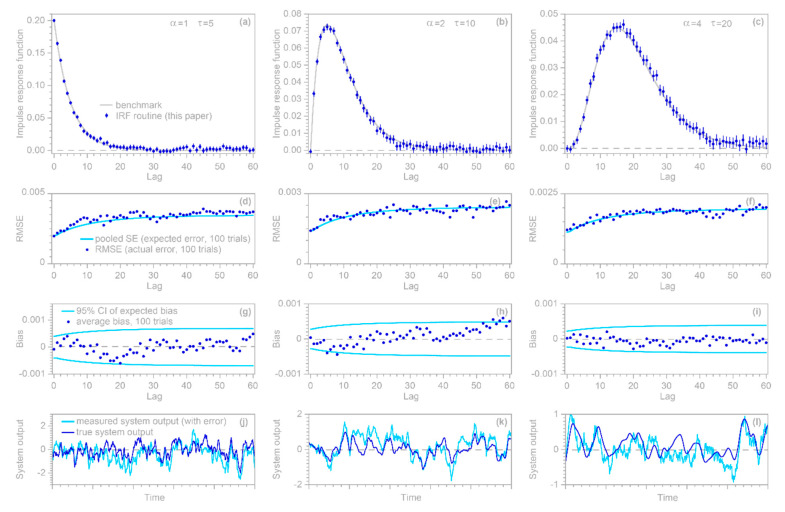
Benchmark tests of impulse response function (IRF) calculations. Top row (**a**–**c**): IRFs calculated from noisy time series (dark symbols), compared to the true convolution kernels (gray lines) used as benchmarks. Error bars indicate one standard error. Benchmarks are gamma distributions with shape factors of 1, 2, and 4 in the left, center, and right columns, respectively. Second row (**d**–**f**): test of standard error estimates, obtained by repeating the analysis in the top row for 100 different randomizations. The estimates of expected deviations from the benchmark (the pooled standard errors at each lag, shown by light blue lines) agree with the actual deviations from the benchmark (root mean square average of deviations from the benchmark at each lag, shown by dark blue symbols). Third row (**g**–**i**): dark blue points show the average deviation from the benchmark in 100 different randomizations of the analysis in the top row, and the light blue lines show the 95% confidence interval of average deviations from the benchmark. The dark points generally lie between the blue lines, indicating that the IRFs are unbiased. Bottom row (**j**–**l**): true convolution output $y_{j}$ (dark blue lines) and the ARMA noise-corrupted $y_{j}$ used as input to the IRF calculations (light blue lines). Five hundred time steps, equaling five percent of each time series, are shown.

**Figure 3 sensors-22-03291-f003:**
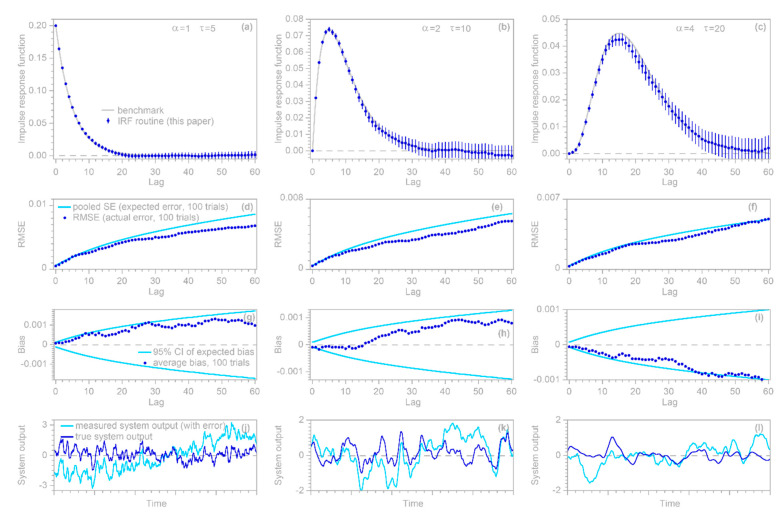
Benchmark tests of impulse response function (IRF) calculations with nonstationary ARIMA noise. Top row (**a**–**c**): IRFs calculated from noisy time series (dark symbols), compared to the true convolution kernels (gray lines) used as benchmarks; error bars indicate one standard error. Benchmarks are gamma distributions with shape factors of 1, 2, and 4 in the left, center, and right columns, respectively. Second row (**d**–**f**): test of standard error estimates, obtained by repeating the analysis in the top row for 100 different randomizations. The estimates of expected deviations from the benchmark (the pooled standard errors at each lag, shown by light blue lines) roughly agree with the actual deviations from the benchmark (root mean square average of deviations from the benchmark at each lag, shown by dark blue symbols). Third row (**g**–**i**): dark blue points show the average deviation from the benchmark in 100 different randomizations of the analysis in the top row, and the light blue lines show the 95% confidence interval of average deviations from the benchmark. The dark points generally lie between the blue lines, indicating that the IRFs are unbiased. Bottom row (**j**–**l**): true convolution output $y_{j}$ (dark blue lines) compared to the ARIMA noise-corrupted $y_{j}$ used as input to IRF calculations (light blue lines). Five hundred time steps, equaling five percent of each time series, are shown. Because the noise is nonstationary, the noise-corrupted time series (light blue lines) eventually wander far away from the true $y_{j}$ (dark blue lines). The bottom panels show time intervals where they nearly overlap, so that they can be visualized together.

**Figure 4 sensors-22-03291-f004:**
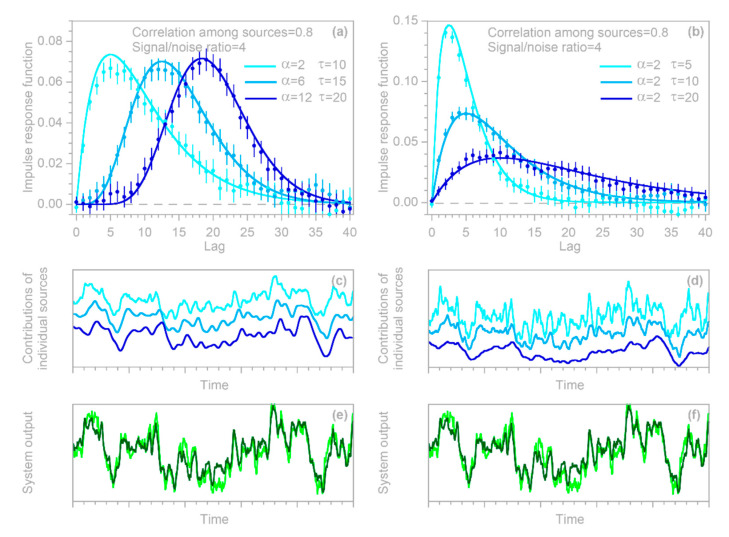
Benchmark test with demixing of impulse response functions from three correlated sources that are mixed together in a system output that is corrupted by ARMA noise. The left column (**a**,**c**,**e**) shows three sources convolved with kernels exhibiting roughly similar dispersion, but different lag times, and the right column (**b**,**d**,**f**) shows three sources convolved with kernels exhibiting different degrees of dispersion. The top panels (**a**,**b**) show the three sources’ benchmark convolution kernels (lines) and their impulse response functions calculated from the individual inputs and their combined output (points). Error bars indicate 1 standard error. The middle panels (**c**,**d**) show each of the three convolved sources’ contributions to the system output, with colors matching the corresponding convolution kernels and IRFs in the top panel, and with each line shifted vertically for clearer visualization. The bottom panels (**e**,**f**) compare the combined system output (the sum of the three curves shown in the middle panels, shown in dark gray), compared to the ARMA noise-corrupted $y_{j}$ used as input to the IRF calculations (shown in light gray). Values of α and τ in (**a**,**b**) are parameters of the gamma distributions used for the benchmark impulse responses. The middle and bottom panels show 500 time steps, equaling five percent of each time series.

**Figure 5 sensors-22-03291-f005:**
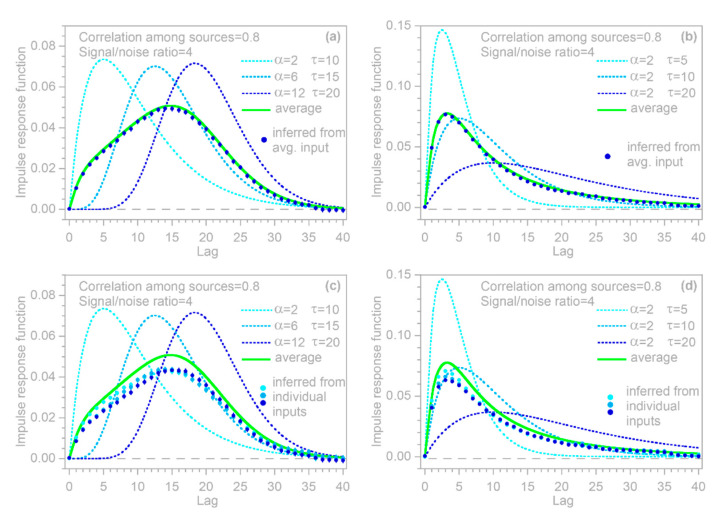
Benchmark test examining whether the average impulse response of the heterogeneous system shown in Figure 4 can be accurately inferred from the average of its three inputs (**a**,**b**), or from each input individually (**c**,**d**). As in Figure 4, the left column (**a**,**c**) shows three sources convolved with kernels exhibiting roughly similar dispersion, but different lag times, and the right column (**b**,**d**) shows three sources convolved with kernels exhibiting different degrees of dispersion. The impulse responses of the three heterogeneous inputs are shown by the thin dotted lines, and their averages (representing the whole-system impulse response) are shown by the thick gray lines. The dots show the IRFs computed from the whole-system output and averaged input (top panels), and from the whole-system output and the individual inputs (bottom panels, with dot colors keyed to match the impulse responses of the corresponding individual inputs). Error bars are shown if they are larger than the plotting symbols. Values of α and τ in (**a**,**b**) are parameters of the gamma distributions used for the benchmark impulse responses.

**Figure 6 sensors-22-03291-f006:**
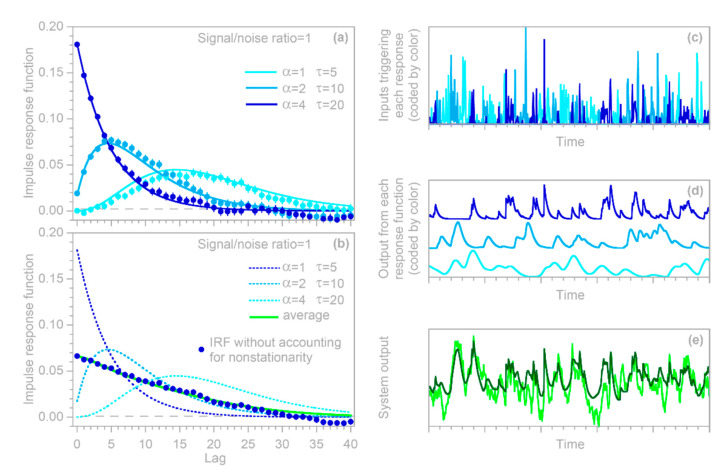
Benchmark test with a nonstationary system that randomly shifts between three different impulse response functions. The three impulse responses are shown by the light, medium, and dark blue curves in (**a**,**b**), and are triggered by the correspondingly colored inputs in (**c**), yielding the output components in (**d**). These output components are not individually identifiable, but instead combine to form the system output in (**e**), shown with (light gray) and without (dark gray) added ARMA noise. The true system output (dark gray) in (**e**) and its individual components in (**d**) are unobservable in practice. The estimated impulse response functions shown by the solid circles in (**a**) are inferred from the input in (**c**) and the ARMA-noise-corrupted output signal (light gray) in (**e**) using the IRF routine. If one treats the system as if it is stationary (by ignoring the differences among the colored input times in (**c**), the IRF routine yields the solid circles shown in (**b**), which correspond closely to the ensemble average (light gray curve) of the three impulse responses (dashed curves). Values of α and τ in (**a**,**b**) are parameters of the gamma distributions used for the benchmark impulse responses. Noise in (**e**) is ARMA(1,2), with an AR coefficient of 0.9 and MA coefficients of −0.2 and +0.2, added to the true system output at a signal-to-noise ratio of 1. Error bars in (**a**,**b**) indicate 1 standard error, and lines in (**d**) are shifted vertically for clearer visualization. Panels (**c**–**e**) show 500 time steps, equaling five percent of each time series.

**Figure 7 sensors-22-03291-f007:**
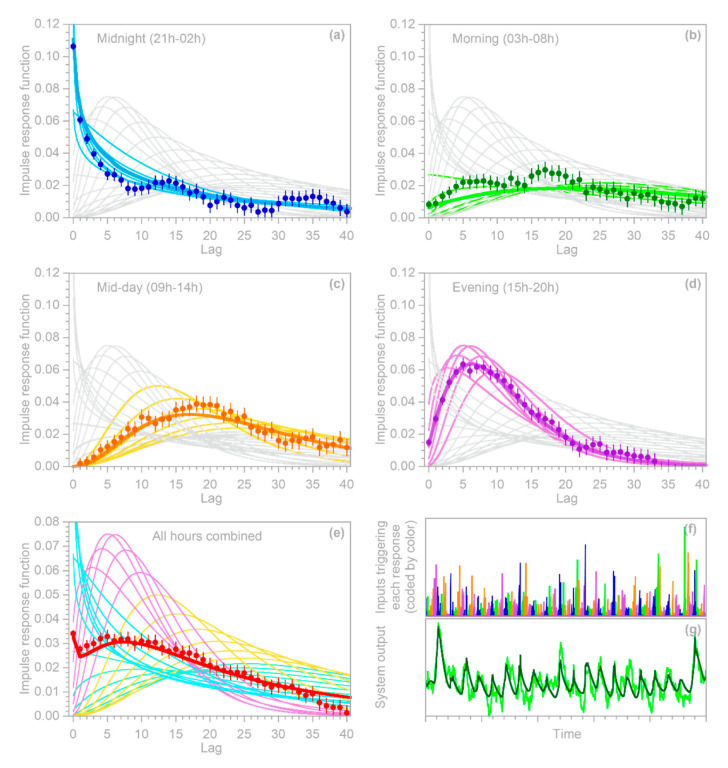
Benchmark test with a nonstationary system whose impulse response varies with time of day. Impulse responses for each hour are shown by the thin lines. These are grouped into four six-hour periods, as shown by the colored lines in (**a**–**d**), with the average impulse responses for each six-hour period shown by the thicker and slightly darker colored lines. The inputs to the benchmark system are random (**f**), but lead to an irregular daily cycle in the output (**g**), owing to the sharper impulse response during night-time (**a**). Using only the input shown in (**f**) and the ARMA-noise-corrupted output shown in light gray in (**g**), the IRF routine yields the solid circles shown in (**a**–**d**), which exhibit broadly similar patterns to the average impulse responses for the corresponding times of day (thicker colored lines). Alternatively, if one treats the system as if it is stationary (by ignoring the differences among the colored input times in (**f**), the IRF routine yields the solid circles shown in (**e**), which correspond closely to the ensemble average (thick red curve) of the hourly impulse responses (thin lines). Error bars in (**a**–**e**) indicate 1 standard error. Noise in (**g**) is ARMA(1,2), with an AR coefficient of 0.9 and MA coefficients of −0.2 and +0.2, added to the true system output at a signal-to-noise ratio of 1. Panels (**f**,**g**) show 500 time steps, equaling five percent of each time series.

**Figure 8 sensors-22-03291-f008:**
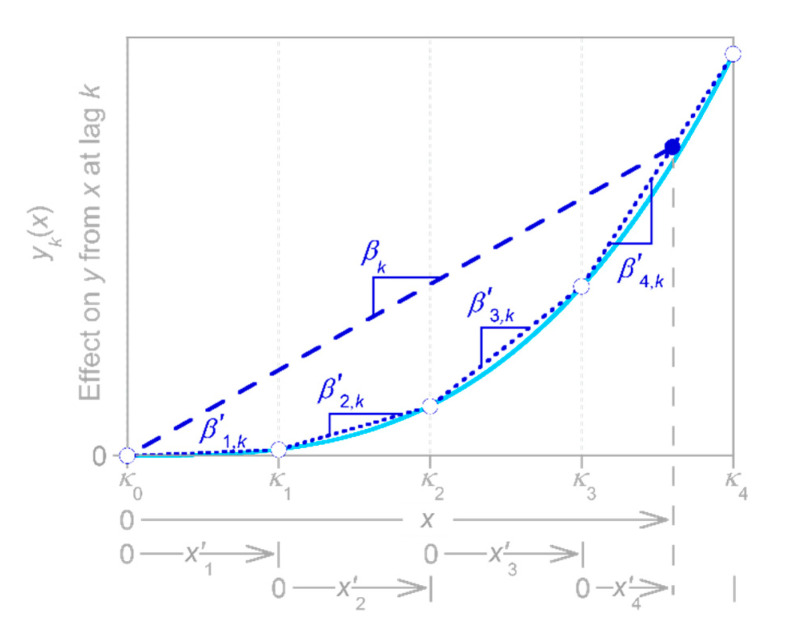
Broken-stick approximation to nonlinear impulse response. Schematic of the (unknown) nonlinear dependence of output y on input x at some lag k, and its approximation by a piecewise linear function defined by knots $\kappa_{\ell }$ and the associated slopes $\beta_{\ell ,k}^{’}$ of each segment between successive knots. The increments$\;x_{\ell }^{’}$ of x within each segment, as illustrated below the x axis, facilitate the estimation of the impulse response coefficient $\beta_{k}$ via broken-stick regression, as described in Equations (41)–(46). For any given value of x, such as the value indicated by the dashed gray line, the system response is approximated by the corresponding point on the piecewise linear curve (the solid blue circle). This response equals the product of the vector of increments $x_{\ell }^{’}$ and the vector of the associated slopes $\beta_{\ell ,k}^{’}$. The impulse response coefficient $\beta_{k}$ is this integral divided by x (i.e., the slope of the dashed blue line). The knots $x=\kappa_{\ell }$ are specified by the user. The corresponding segment slopes $\beta_{\ell ,k}^{’}$, determined by fitting Equation (44 ) to time series of x and y, can be used to infer the associated values of $y_{k}\left(x=\kappa_{\ell }\right)$ and thus to estimate the nonlinear dependence of y on x at each lag k.

**Figure 9 sensors-22-03291-f009:**
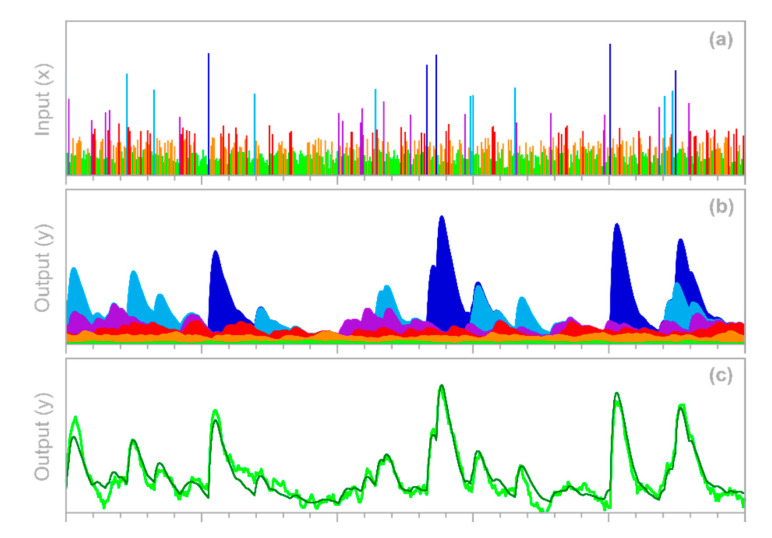
Example input and output time series for nonlinear deconvolution benchmark tests. The input is a log-normally distributed white noise time series, shown in (**a**) with different colors representing different ranges of values. Each value in (**a**) was convolved with a gamma distribution with shape factor of 2 and mean lifetime of 10, but with an amplitude proportional to the square of the input (thus yielding a system response that is proportional to the cube of the input). The resulting true output is shown in (**b**), with the same color coding as in (**a**) indicating the ranges of input values. The more numerous low-intensity inputs (e.g., gray and orange) in (**a**) result in a relatively stable base output in (**b**), with the rarer but higher-intensity inputs (e.g., purple, light blue, and dark blue) generating markedly larger but more intermittent outputs. The combined true system output, shown by the dark gray line in panel (**c**), was then corrupted by ARMA noise at a signal-to-noise ratio of 4, to yield the apparent system output shown by the light gray line in panel (**c**). The plots shown here comprise 500 time steps, or 5% of the time series used for the benchmark tests.

**Figure 10 sensors-22-03291-f010:**
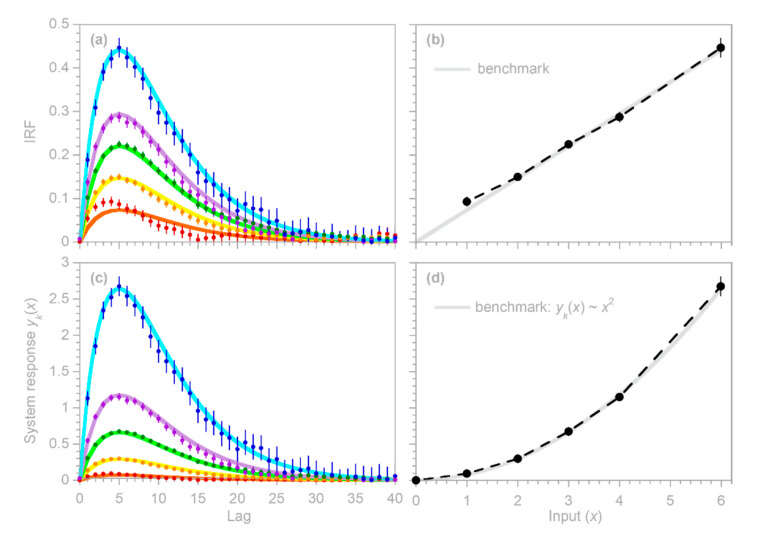
Benchmark test of nonlinear deconvolution. Benchmarks (lines) and estimated impulse responses (dots) are shown for a benchmark test in which the impulse response function (**a**,**b**) is proportional to the input, and thus the system’s response—the impulse response function times the input (**c**,**d**)—is a quadratic function of the input. The impulse response functions (system response per unit input) and the system responses themselves, along with their benchmarks (**a**,**c**), are evaluated at the knots $\kappa_{\ell }$ (in this case, at values of 1, 2, 3, 4, and 6, the highest value of the input x). The benchmarks are shown by the colored lines in panels (**a**,**c**), with the impulse response functions shown by the corresponding colored symbols. The corresponding peak values of the impulse response function and system response are shown in panels (**b**,**d**), with the benchmark relationships shown by gray lines. Error bars show 1 standard error, wherever this is larger than the plotting symbols.

**Figure 11 sensors-22-03291-f011:**
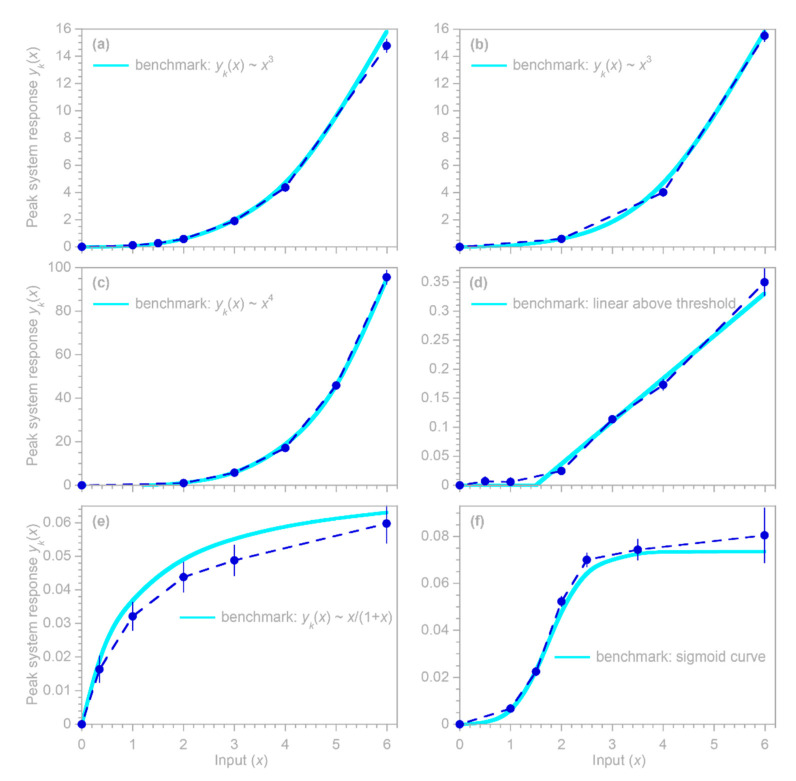
Benchmarks (light blue lines) and estimated peak system responses (dark blue dots) for benchmark tests featuring a range of nonlinear relationships between the peak system response $y_{k}\left(x\right)$ and the input x. In both (**a**,**b**), the benchmark’s peak response varies as the cube of the input, but knots divide the input values into six intervals in (**a**) and only three intervals in (**b**). The other relationships include a quartic curve (**c**), a linear function above a threshold (**d**), the Michaelis–Menten saturation curve (**e**), and a sigmoid curve (**f**). Except for some systematic bias in (**e**), the estimates derived from the approach outlined in Section 5.1 (blue circles) closely approximate the benchmark curves, with deviations similar to the estimated standard errors (as shown by the error bars, where these are larger than the plotting symbols).

**Figure 12 sensors-22-03291-f012:**
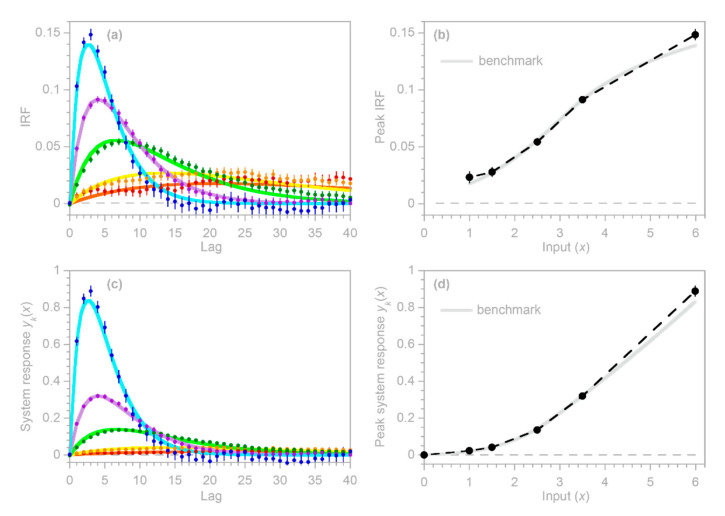
Nonlinear deconvolution of a benchmark impulse response that changes its mean lifetime as a function of the input. Benchmarks (lines) and estimated impulse responses (dots) are shown for a test in which the gamma-distributed benchmark impulse response function’s mean lifetime decreases from roughly 40 at the lowest knot (corresponding to the red curve and dots) to approximately 5 at the highest knot (corresponding to the blue curve and dots). The impulse response functions (system response per unit input) and the system responses themselves, along with their benchmarks (**a**,**c**), are evaluated at the knots $x=\kappa_{\ell }$ (in this case, at values of 1, 1.5, 2.5, 3.5, and 6, the highest value of the input x ), indicated by the colors red, orange, gray, purple, and blue, respectively. The corresponding peak values of the impulse response function and system response are shown in panels (**b**,**d**). Error bars show 1 standard error, wherever this is larger than the plotting symbols.

**Figure 13 sensors-22-03291-f013:**
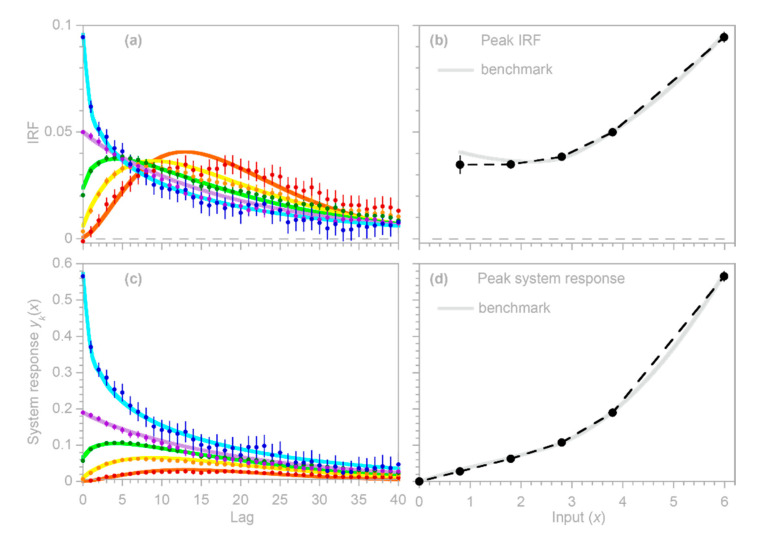
Nonlinear deconvolution of a benchmark impulse response that changes its shape as a function of the input. Benchmarks (lines) and estimated impulse responses (dots) are shown for a test in which the gamma-distributed benchmark impulse response function’s shape factor α decreases from roughly 3 at the lowest knot (corresponding to the red curve and dots) to approximately 0.7 at the highest knot (corresponding to the blue curve and dots). The impulse response functions (system response per unit input) and the system responses themselves, along with their benchmarks (**a**,**c**), are evaluated at the knots $x=\kappa_{\ell }$ (in this case, at values of 0.8, 1.8, 2.8, 3.8, and 6, the highest value of the input x ), indicated by the colors red, orange, gray, purple, and blue, respectively. The corresponding peak values of the impulse response function and system response are shown in panels (**b**,**d**). Error bars show 1 standard error, wherever this is larger than the plotting symbols.

**Figure 14 sensors-22-03291-f014:**
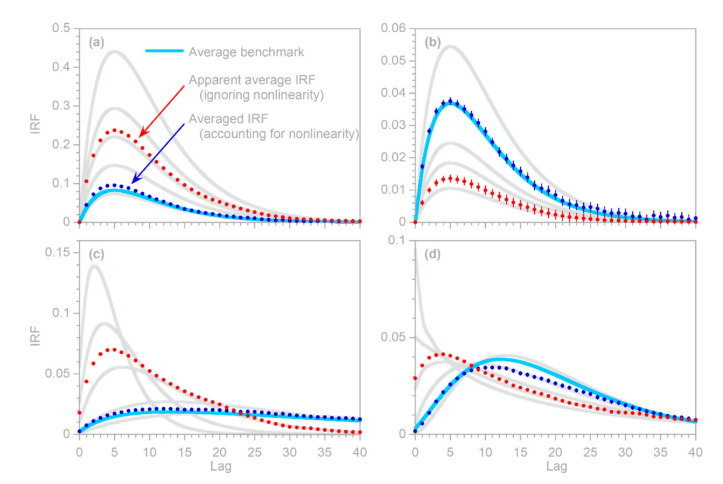
Bias in apparent average impulse response in four nonlinear benchmark systems. Benchmarks include (**a**) quadratic relationship between input and output (Figure 10); (**b**) Michaelis–Menten relationship between input and output (Figure 11e); (**c**) gamma mean lifetime decreases with increasing input (Figure 12); (**d**) gamma shape factor decreases with increasing input (Figure 13). Test procedures are identical to Figure 10, Figure 11, Figure 12 and Figure 13, except signal-to-noise ratio is 100 so that mean IRFs are shown with minimal scatter. Gray lines show benchmark impulse responses at knot values $x=\kappa_{\ell }$; these are not evenly spaced either in x or in percentiles of x, and thus the spacing between the gray lines does not indicate the underlying degree of nonlinearity. Light blue lines show the average benchmark impulse response functions for each system, averaged over all values of x. Dark blue points show the average impulse response functions estimated from Equations (43) and (47). These estimates generally follow the average benchmarks. Dark red points show naïve estimates of impulse response functions estimated using the methods of Section 2.1, Section 2.2, Section 2.3 and Section 2.4, without accounting for the systems’ nonlinear behavior. These naïve estimates can be much larger (**a**) or smaller (**b**) than the benchmark average impulse response functions shown by the blue lines, or have a substantially different shape (**c**,**d**). Error bars show 1 standard error, wherever this is larger than the plotting symbols.

**Figure 15 sensors-22-03291-f015:**
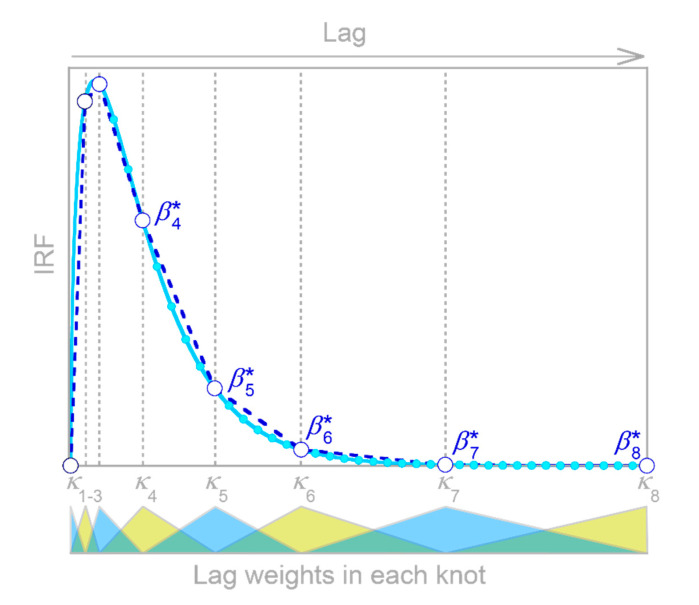
Piecewise linear approximation (dashed dark blue line) to an impulse response function (light blue line), with 40 conventional IRF coefficients (light blue dots) replaced by values at 8 knots (open circles). The overlapping yellow and blue triangles depict the relative influence of each lag on the even and odd numbered knots, respectively (see Equations (51) and (52)).

**Figure 16 sensors-22-03291-f016:**
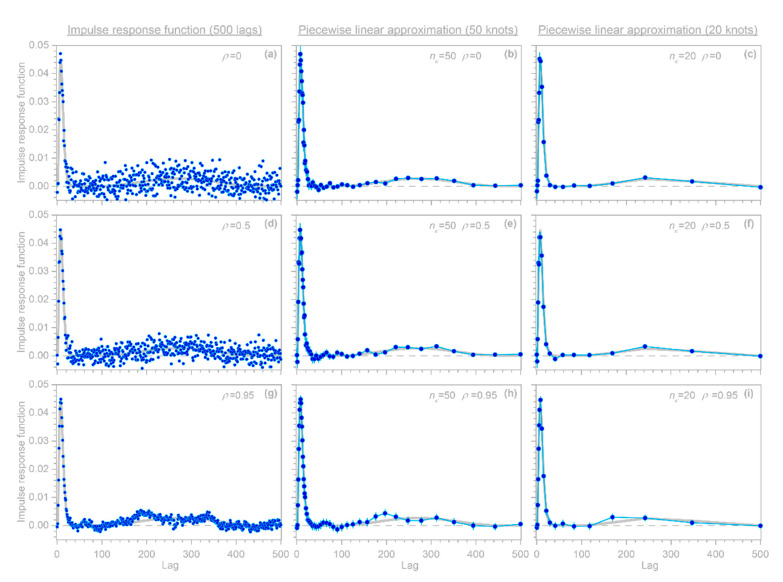
Benchmark test of the piecewise linear approach to estimating multi-scale impulse response functions. The benchmark IRF (gray line) is a gamma distribution with shape factor of 4, combined with a broad, low Gaussian curve at lags between roughly 90 and 450. The gray dashed line indicates zero on the vertical axis. The left column (**a**,**d**,**g**) shows conventional impulse response functions estimated for the first 500 lags (blue dots, shown without error bars). The middle and right columns (**b**,**c**,**e**,**f**,**h**,**i**) show piecewise linear IRFs obtained by the approach outlined in Section 6.1 (with 50 and 20 knots, respectively), with error bars wherever these are larger than the plotting symbols. The three rows show different degrees of autocorrelation in the added Gaussian noise, from white noise (first-order serial correlation coefficient ρ=0, top row), to moderately autocorrelated noise (ρ=0.5, middle row), and strongly autocorrelated noise (ρ=0.95, bottom row); in all cases, the signal-to-noise ratio is 1. The brief, sharp system response is clearly detected in all cases shown here. However, the weaker, longer-term response is obscured (**a**,**d**) or distorted (**g**) in the conventional IRF, and is only revealed by the piecewise linear approach (middle and right columns). As the noise becomes more serially correlated (middle and bottom rows), more widely spaced knots (right column) are needed to accurately reflect the long-tail behavior of the benchmark impulse response.

## Data Availability

R scripts that implement the techniques presented here and perform the benchmark tests in Figure 2, Figure 3, Figure 4, Figure 5, Figure 6, Figure 7, Figure 8, Figure 9, Figure 10, Figure 11, Figure 12, Figure 13, Figure 14, Figure 15 and Figure 16 are available as Appendix A accompanying this article. The same scripts are also available at Kirchner, James W. (2022). Impulse response functions for nonlinear nonstationary and heterogeneous systems. EnviDat. https://doi.org/10.16904/envidat.312, which will be updated with bug fixes as needed.

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
