# Peer review of "Impulse Response Functions for Nonlinear, Nonstationary, and Heterogeneous Systems, Estimated by Deconvolution and Demixing of Noisy Time Series"

_sensors, 2022, doi:10.3390/s22093291_

Round 1
Reviewer 1 Report
1. The paper introduces some parameter estimation methods about typical AR, MA and ARMA models in system identification from input-output data, Especially (finite) impulse response model identification. In fact, there are many methods about more complex models in system identification such as CAR (C-controlled), CARMA, CARAR, CARARMA. They include AR, MA and ARMA models as special cases. Therefore some related original works about the parameter estimation of ARMA models should be mentioned in the paper, such as [1] Performance analysis of estimation algorithms of non-stationary ARMA processes, IEEE Trans. Signal Process. 2006, 54(3): 1041-1053. [2] Identification of Hammerstein nonlinear ARMAX systems, Automatica, 2005, 41(9): 1479-1489. [3] Hierarchical gradient based and hierarchical least squares based iterative parameter identification for CARARMA systems, Signal Processing, 2014, 97: 31-39. [4] Two-stage least squares based iterative estimation algorithm for CARARMA system modelingm Applied Mathematical Modelling, 2013, 37(7): 4798-4808.
2. Moreover, the Impulse response functions or Finite Impulse Response (FIR) model are a simplest model. In fact, the parameters of the transfer fiunctions can be estimated through impulse responses, thus the following related contributions about impulse responses should be surveyed and mentioned in the paper such as [1] Separable Newton recursive estimation method through system responses based on dynamically discrete measurements with increasing data length. International Journal of Control Automation and Systems, 2022, 20(2): 432-443. [2] Decomposition strategy-based hierarchical least mean square algorithm for control systems from the impulse responses. International Journal of Systems Science, 2021, 52(9): 1806-1821.
[3] Separable multi-innovation stochastic gradient estimation algorithm for the nonlinear dynamic responses of systems. International Journal of Adaptive Control and Signal Processing, 2020, 34(7): 937-954.
[4] Separable recursive gradient algorithm for dynamical systems based on the impulse response signals. International Journal of Control Automation and Systems, 2020, 18(12): 3167-3177.
3. English is acceptable.
3. The paper is acceptable if the author addresses the above comments.
Author Response
Thanks for these comments.
Of the eight papers mentioned by the reviewer, Feng Ding is the first author on four of them and a co-author on three of the others. The first author of the other four papers is Ling Xu, who appears to be a former student of Prof. Ding. It is surprising that all eight papers mentioned by the reviewer seem to come from one research group. This is the first time in my career that I have seen such a thing.
Concerning the reviewer's first point, we should recognize that the main focus of this manuscript is quite different from the conventional literature on AR, MA, and ARMA models (as well as further elaborations of these, including CARARMA etc.). As explained in the last paragraph of Section 2.2, typical system identification papers usually assume that the ARMA behavior is inherent in the system itself, and the main objective is to estimate the coefficients that describe that behavior. Here, instead, the ARMA behavior is assumed to be part of the noise (not the system), and the main objective is to determine the impulse response linking the output y(t) to the input x(t). That is, we care about the "X" part of an ARX model, not the "AR" part. And because the "AR" part is assumed to be noise, and not driven by x(t), the input is not informative for estimating the AR coefficients. Importantly, because the AR/ARMA/ARIMA behavior is in the noise, not the system, obtaining the impulse response function requires deconvolving the fitted coefficients, as described in Sections 2.3 and 2.6. The approach is different when the AR/ARMA/ARIMA behavior is in the system itself, as in the four papers mentioned in the reviewer's comment. The iterative/recursive techniques described in those papers often require hundreds or thousands of data points, and dozens or hundreds iterations to converge, even for simple problems with 4-6 parameters and very low noise (and when the correct ARMA orders are known in advance). By contrast, the methods developed here can quantify dozens, hundreds, or even thousands of parameters, and their uncertainties, in a single iteration (or a small handful of iterations if one wants to perform robust estimation or search for an optimal order h for AR correction).
Regarding the reviewer's second point: yes, parameters of transfer functions can be estimated from impulse responses, *if* we actually have an ARMA system and *if* that system's parameters are stable over time. In the more general case, this will not be true, and we can have (potentially nonstationary) IRF coefficients but no straightforward way to map them to ARMA coefficients, or those coefficients may not be meaningful if, for example, the system is not really an ARMA system in the first place. However, by parameterizing the impulse response as a series of exponential functions, one creates a model that is nonlinear in the parameters and thus requires complex and inefficient solution methods. For example, in the reviewer's ref. [4], many iterations are apparently required to achieve reasonable convergence – in a test case where one already knows both the form and the order of the model that must be fitted, and where the level of noise is very low (the signal-to-noise ratio appears to be approximately 100). In the reviewer's ref. [1], the level of noise is even lower (the signal-to-noise ratios apparently range from 625 to 10,000).
_________________________________________
I have expanded the discussion at the end of Section 2.2 to point out the existence of CARARMA models and to distinguish them from the approach presented here:
"The form of Equation (14) is similar to a conventional SISO (Single Input, Single Output) ARX (Autoregressive with eXogenous variables) model {e.g.`, \Ljung, 1999 #2526; Tangirala, 2015 #2523}, but there are three essential differences. The first difference is that in ARX models, the focus is usually on the autoregressive part, and the objective is usually to be able to make one-step-ahead forecasts of the next value of y_j, based mostly on y's relationship to its own prior values. By contrast, in the analysis presented here, the focus is on the exogenous variable x and its lags, and on estimating their structural relationship to y.
The second difference is that in ARX models, the autoregressive terms Ï•_1 y_(j-1), Ï•_2 y_(j-2), etc. describe autoregressive behavior in the system itself (an "equation error model"), rather than correcting for autoregressive noise in the error term (an "output error model"). (Although these two model types can be combined in so-called CARARMA models, for which iterative and hierarchical estimation algorithms have been proposed {e.g.`, \Ding, 2014 #2612}, such complex models need not concern us here because in the present analysis only the noise is assumed to be autoregressive.) Because it attributes autoregressive behavior to y_j rather than to ε_j, an ARX model evaluates the b coefficients only for lags from 0 to m rather than from 0 to m+h as shown in Equation (14). This distinction is important because without the extra coefficients b_(m+1)…b_h, Equation (14) would not be the same as Equation (9) and thus a solution for Equation (14) would not be a solution for the original problem as specified by Equation (4) combined with Equation (6).
The third crucial difference is that in an ARX model, the b coefficients would directly measure the effects of the (lagged) external forcing x_(j-k). Here, by contrast, these effects are measured by the β coefficients, which must be deconvolved from the b coefficients as described in the next section."
_________________________________________
Near the end of Section 5.2, I have added the following paragraph, to point out how my nonlinear analysis differs from Hammerstein systems:
"The nonlinearities considered here bear a superficial resemblance to Hammerstein systems, for which several identification algorithms have been proposed (e.g., {Greblicki, 2002 #2610; Ding, 2005 #2611}), but there are two important differences. First, Hammerstein systems consist of two parts connected in series: the input is first transformed by a nonlinear function, which in turn drives a linear AR or ARMA system. Thus the entire system response scales nonlinearly as a function of the input. In systems like Equations (41) and (42), by contrast, the system response may vary in shape, not just in scale, as the input changes. Second, in Hammerstein systems, the autoregressive or ARMA behavior is assumed to originate within the system itself, whereas here it is assumed to characterize the noise. Thus the primary focus here is on estimating the impulse response coefficients and their nonlinear dependence on the input x, rather than the AR coefficients Ï•. This has important implications for the solution method, as described at the end of Section 2.2 above."
_________________________________________
Just before Section 6.1, I have added a reference to Xu (2020) and changed the surrounding discussion slightly to read:
"Another common approach to this problem is to assume that the IRF has a known functional form (e.g., a gamma distribution or a set of exponentials), and to estimate the parameters for that function. However, this typically results in a model that is nonlinear in its parameters, with the implication that parameters must be estimated by computationally intensive iterative search methods { e.g.`, \Xu, 2020 #2613} and one can never be certain that the single best set of parameters has been found (the local op-timum problem). Such approaches also depend critically on the assumption that the chosen function is the correct one, which is usually impossible to verify."
Reviewer 2 Report
This paper is off topic for Sensors journal.
My recommendation is to reject this paper and encourage to send it to another journal in the field of signal processing and control.
Author Response
Presumably the question of whether a manuscript fits within the scope of the journal is first evaluated by the editors, before it is sent out for review. (At least that has been the case throughout the more than 40 years that I have been involved in academic publishing, as an author, reviewer, and associate editor.)
The scope of Sensors, as declared at https://www.mdpi.com/journal/sensors/about, specifically includes "Signal processing, data fusion and deep learning in sensor systems", as well as "Machine/deep learning and artificial intelligence in sensing and imaging" and "Communications and signal processing". The present manuscript clearly fits within the first of these topics, but is potentially consistent with the other two as well.
If the editors don't want this paper in Sensors, then that is their decision to make; but in that case they normally would have made that decision before asking reviewers to spend time on it.
Reviewer 3 Report
The paper proposes a method to estimate the impulse response function of a system from its input-output data. The method is demonstrated on nonlinear, nonstationary, and heterogeneous systems to estimate the IRF utilizing simulated benchmark datasets. The novelty of the proposed methodology is evident, and the paper is well written. The paper can be considered for publication after the author addresses the following minor comments.
- In the general case of nonlinear systems, it is possible to have more than one stable response, quasi-periodic response, or even chaotic response. How will the proposed method of estimating IRF behave in such nonlinear regimes? The author is suggested to mention the limitations of the proposed method, if any, in the discussion section.
- How does the accuracy of IRF estimation depend on the auto-correlation function of the input? The author is suggested to comment on whether the variance of the estimates is lower for broadband input compared to a narrowband input.
Author Response
Thanks a lot for these comments.
The first point, concerning nonlinear systems, illustrates the need for a clarifying comment in the discussion. Specifically, when the manuscript refers to "nonlinear systems" it means, as specified in the abstract, "systems whose different amplitudes and shapes of impulse responses, reflecting different input intensities, are overprinted on one another in the output." In other words, it means systems that are fundamentally convolutions (as in Equation 1), but that depend nonlinearly on the lagged inputs (as in Equations 41 and 42). But of course the term "nonlinear systems" is also widely used as a shorthand for nonlinear *dynamical* systems, in which the derivatives controlling the time evolution of the system are nonlinear functions of the system state itself. Such nonlinear dynamical systems can exhibit bifurcations and chaotic dynamics, and as such, they do not have impulse response functions. So one can say that the approach presented here cannot be expected to reliably estimate impulse response functions in systems where those impulse response functions do not exist.
To address this point, I have added the following text to the discussion:
"…the term "nonlinear" as used here refers to systems whose impulse response functions depend nonlinearly on the input intensity, and thus can be quantified by nonlinear deconvolution techniques like those described in Section 5. Similar terminology is of-ten used to refer to a broad class of nonlinear dynamical systems – that is, systems of nonlinear differential equations characterized by bifurcations and chaotic dynamics. Such systems are not described by linear convolutions like Equation (1), or nonlinear convolutions like Equations (41) and (42), and thus cannot be deconvolved by the methods outlined here. Simply put, such systems do not have impulse response func-tions, so the methods outlined here cannot estimate them."
The second point is also important. I have added a paragraph to the discussion explaining that:
"Readers should keep in mind that the reliability of any deconvolution method, including the methods described here, will depend on the autocorrelation behavior (and thus the frequency content) of the input to the system. In convolutional systems like Equation (1), the impulse response will be poorly constrained at any frequency for which the input to the system exhibits little or no variation. This is inherent in the mathematics of convolution, and is independent of the particular deconvolution methods that are used. The principles can be easily seen by re-casting the linear convolution in Equation (1) as its Fourier transform:
Y(f)=B(f)∙X(f) (54)
where Y(f), B(f), and X(f) are the (complex) Fourier transforms of y(t), β(t-τ), and x(t-τ). From Equation (54) one can see that Y(f) will have no spectral power at any frequency f where X(f) contains no spectral power. Thus the deconvolution estimate of B(f),
B(f)=Y(f)⁄X(f) (55)
will be undefined at that frequency, because Equation (55) will be dividing zero by zero. For nonlinear convolutions the situation is more complex, because an input X(f) at any given frequency f will yield outputs not only at f but also at its harmonics, and thus Equations (54)-(55) will no longer apply.
Nonetheless, the principle remains that broadband inputs will generally yield more reliable estimates of the impulse response function than narrowband inputs will. In most of the benchmark tests shown here, the inputs are Gaussian white noise (and hence are ideal broadband signals), except for Figures 2, 3, and 16, in which the inputs are Gaussian noise with a lag-1 correlation of 0.5. Fortunately, many real-world systems also have broadband inputs, and thus may be well suited to deconvolution approaches like those outlined here. Users with any concerns in this regard are encouraged to explore them with benchmark tests tailored to the characteristics of their own systems and data sets. "
Round 2
Reviewer 2 Report
Dear Author,
I have to redo my reject recommendation.
It is of course the Editor's duty to take a reviewer decision into consideration or not (at least this has been the case during the more than 20 years that I have been involved in academic publishing, as an author, reviewer, academic editor, associate editor and editor-in-chief).